# Environmental exposures associated with the gut microbiome and resistome of pregnant women and children in Northwest Ecuador

Published online:

Irmarie Cotto [1], Viviana Albán [1,15], Ana Durán-Viseras [2,15], Kelsey J. Jesser[1], Nicolette A. Zhou[1], Caitlin Hemlock[1], April M. Ballard [3], Christine S. Fagnant-Sperati [1], Gwenyth O. Lee[4,5], Janet K. Hatt [2], Charlotte J. Royer[2,6], Joseph N. S. Eisenberg[7], Gabriel Trueba [8], Konstantinos T. Konstantinidis[2], Karen Levy [1] ✉, Erica R. Fuhrmeister [1,9] ✉ On behalf of the ECoMiD Authorship Group*

Inadequate water, sanitation, and hygiene (WASH) infrastructure may increase exposure to antimicrobial resistance (AMR). In addition, close human-animal interactions and unregulated antibiotic use in livestock facilitate the spread of resistant bacteria. We use metagenomic sequence data and multivariate models to assess how animal exposure and WASH conditions affect the gut resistome and microbiome in 53 pregnant women and 84 children in Ecuador. Here we show improving WASH infrastructure and managing animal exposure may be important in reducing AMR but could also reduce taxonomic diversity in the gut. *Escherichia coli*, *Klebsiella pneumoniae*, and clinically relevant antimicrobial resistance genes (ARGs) are detected across all age groups, but the highest abundance is found in children compared to mothers. In mothers, higher animal exposure trends towards a higher number of unique ARGs compared to low animal exposure and is significantly associated with greater taxonomic diversity. In addition, mothers with sewer systems or septic tanks and piped drinking water have fewer unique ARGs compared to those without, and mothers with longer duration of drinking water access have lower total ARG abundance. In contrast, few associations are observed in children, likely due to the dynamic nature of the gut microbiome during early childhood.

Antimicrobial resistance (AMR) is a significant global public health issue, particularly affecting low, lower-middle, and upper-middle-income countries (LMICs)[1,2]. Inadequate water, sanitation, and hygiene (WASH) services, restricted access to healthcare resources, and lack of regulatory frameworks for antimicrobial use contribute to the amplified burden of AMR in these regions[3]. Inconsistent access to clean water and adequate sanitation infrastructure can lead to increased exposure to antimicrobial residues, antimicrobial-resistant bacteria, and antimicrobial resistance genes (ARGs). Inadequate WASH also leads to an increased infectious disease burden and higher demand for antimicrobials[4]. The World Health Organization (WHO) has emphasized the urgent need for

---

A full list of affiliations appears at the end of the paper. *Lists of authors and their affiliations appear at the end of the paper. ✉e-mail: klevyx@uw.edu; efuhrm@uw.edu

comprehensive strategies to address AMR, noting disproportionate impacts in LMICs[5].

Close contact with animals for household food production or income, as pets or strays, and in markets can increase microbial exchange between humans and animals in LMICs[6–8]. Extensive use of antibiotics in food animals contributes to the development and spread of resistant bacteria and ARGs[9]. Research assessing global trends in antimicrobial resistance has found a high prevalence of ARGs in livestock, particularly those conferring resistance to tetracyclines and beta-lactams[10], and studies have also identified ARGs in pets[11–13]. In LMICs, the use of antibiotics in livestock is often less regulated compared to high-income countries (HICs), leading to higher rates of AMR[10,14]. In addition, the burden of AMR in animals may be influenced by limited access to veterinary care[1]. Studies in rural Ecuador have documented a diverse array of ARGs in small-scale poultry farms[15–17] and domestic dogs[18–20], with a high prevalence of extended-spectrum beta-lactamase (ESBL) and *mcr-1* producing *E. coli*[21,22]. β-lactamases, particularly ESBLs, have been identified as major drivers of resistance in animals in Ecuador, representing some of the most clinically relevant ARGs circulating in the region[23–25]. Recent metagenomic and genomic studies demonstrate that β-lactamase ARGs such as CTX-M are not only widespread but also frequently associated with mobile genetic elements, increasing their potential for horizontal gene transfer (HGT) across hosts and environments[24–26]. Also, research has shown a dynamic collection of horizontally transferred ARGs and mobile genetic elements (MGEs) shared between the microbiomes of children and domestic animals (i.e., dogs and chickens) in semirural Ecuadorian communities[24], alongside detection of enteric pathogens in both groups[27]. These findings underscore the need for detailed investigation of β-lactam resistance in community settings, where human–animal–environmental interfaces may accelerate the exchange of resistant organisms. While existing studies show an overall trend, location-specific decoupling of WASH characteristics and their impact on AMR is needed to implement site-specific interventions. There is also a need to understand the interplay between animal exposure and WASH conditions and AMR.

Young children are particularly at risk of exposure and vulnerable to the impacts of AMR. Behaviors such as mouthing objects and playing on the ground increase their risk of exposure to microorganisms, including those carrying antimicrobial resistance elements[28]. Moreover, early childhood is a critical period for microbiome development, significantly influencing growth, cognitive function, and immune response[29]. Women, especially in LMICs, also face a high risk of AMR due to biological factors like menstruation, pregnancy, and childbirth, which increase susceptibility to infections and may require antibiotic treatments[30]. The use of unsafe sanitary products and poor sanitation during menstruation can increase infection risks, particularly urinary tract infections[31]. During pregnancy, the need for antibiotics may be increased due to limited access to maternal healthcare and complications during childbirth[32]. In addition, traditional caregiving roles (e.g., healthcare work, child and elderly care and food preparation) expose women to pathogens, further elevating their risk of infection and subsequent antibiotic use[30,33].

A group of highly virulent pathogens of public health importance, *Enterococcus faecium, Staphylococcus aureus, Klebsiella pneumoniae, Acinetobacter baumannii, Pseudomonas aeruginosa, Enterobacter* spp., *and Escherichia coli* (ESKAPEE pathogens) can develop resistance from ARGs circulating in the environment[34]. The genome plasticity of these bacteria facilitates the acquisition and dissemination of ARGs through HGT mechanisms, often mediated by mobile genetic elements like plasmids and transposons[35]. This adaptability enables ESKAPEE pathogens to rapidly develop resistance to multiple antibiotics, complicating treatment options and contributing to the global health threat posed by antimicrobial resistance. For instance, studies have identified the presence of various β-lactamase genes (e.g., $bla_{TEM}$,

$bla_{SHV}$, $bla_{KPC}$) in *K. pneumoniae* and other gram-negative bacteria[36]. Understanding the relationship between ARGs and ESKAPEE pathogens is crucial for developing effective surveillance, prevention, and treatment strategies to combat the escalating challenge of antimicrobial resistance.

While ESKAPEE pathogens are, to varying degrees, culturable, many other bacteria that harbor ARGs and could transfer them to ESKAPEE pathogens are not cultivable[37,38]. Many investigations of AMR focus on traditional culture techniques[39], which do not capture the full breadth of microbial diversity. Culture-based methods may neglect a significant portion of microbial diversity, resulting in an incomplete understanding of microbial communities and their associated AMR potential[40,41]. By directly extracting and sequencing genetic material, metagenomics allows for comprehensive profiling of microbial communities, including unculturable bacteria[42]. This approach enables the identification of novel or previously undetected ARGs, providing a more complete description of the resistome (i.e., the collection of all ARGs present in both pathogenic and non-pathogenic microbes within a given environment) beyond clinically recognized pathogens[43].

Despite growing recognition of the role that environmental factors play in AMR, most existing studies evaluate WASH and animal exposure as broad, composite indicators and rely heavily on culture-based methods or targeted gene detection. In addition, few studies have assessed the interactive effects of individual WASH components and animal exposure on the gut resistome in vulnerable populations such as pregnant women and young children. Our study addresses this gap by using shotgun metagenomic sequencing to investigate how specific conditions (e.g., sanitation type, drinking water availability, and animal exposure) individually and jointly shape the resistome and microbiome in a low-resource setting in Northwest Ecuador.

Here, we (1) investigate the association of environmental factors, specifically, animal exposure, drinking water availability, drinking water source type, and sanitation systems on the gut resistome (i.e., alpha diversity and abundance of clinically relevant ARGs) and microbiome composition (i.e., Nonpareil sequence diversity and the relative abundances of the ESKAPEE pathogens *E. coli* and *K. pneumoniae*); (2) explore whether drinking water and sanitation conditions modify the relationship between animal exposure and the outcomes; and (3) examine the distribution of clinically relevant ARGs, their individual association with animal exposure, their genomic location (chromosomes or plasmids), and their linkage to ESKAPEE pathogens, with particular emphasis on beta-lactam resistance genes. This research is conducted as part of the ECoMiD (*Enteropatógenos, Crecimiento, Microbioma, y Diarrea*) study[44] in Northwest Ecuador, in a region characterized by significant variability in WASH infrastructure and environmental exposures[45].

## Results

### Overview of households included in this study and environmental exposures

Of the 84 households included in this analysis, 32 used piped water (38.1%) as their drinking water source. Other sources included bottled water, rainwater, surface water, protected wells, unprotected wells, and "other" sources. Most of the sanitation systems in the households were toilets that discharge into sewer systems (25.0%) or toilets that discharge to septic tanks (46.4%). Other sanitation systems included toilets that discharge to a pit latrine, improved and ventilated pit latrines, pit latrines with a slab, pit latrines without a slab and "other". Among the 53 mothers and 84 children in our study, 39.6% of mothers and 28.6% of children resided in households equipped with both contained sewage systems and piped water (CS&PW). Regarding drinking water supply, 34.0% of mothers and 23.8% of children resided in households with piped water available 7 days a week, while 26.4% of mothers and 40.0% of children had no piped water. Among the mothers, 47.2% had zero animal exposure, and 26.4% were classified as

## Table 1 | Environmental exposure variables among mothers and children

| | Mothers | Children |
|---|---|---|
| Households | 53 | 84 |
| Samples | 53 | 153 |
| **Sanitation** | | |
| Sewage containment[a] | 41/53 (77.4%) | 60/84 (71.4%) |
| No sewage containment | 10/53 (18.9%) | 19/84 (22.6%) |
| Not available (NA) | 2/53 (3.8%) | 5/84 (6.0%) |
| **Drinking Water** | | |
| Piped water | 27/53 (50.9%) | 32/84 (38.1%) |
| Not piped water | 26/53 (49.1%) | 52/84 (61.9%) |
| **Combined Drinking Water & Sanitation** | | |
| Contained Sewage & Piped Water (CS&PW)[b] | 21/53 (39.6%) | 24/84 (28.6%) |
| Other | 30/53 (53.6%) | 58/84 (69.0%) |
| NA | 2/53 (3.8%) | 2/84 (2.4%) |
| **Animal exposure** | | |
| High | 4/53 (7.5%) | 29/153 (18.9%) |
| Medium | 10/53 (18.9%) | 31/153 (20.3%) |
| Low | 14/53 (26.4%) | 32/153 (20.9%) |
| Zero | 25/53 (47.2%) | 58/153 (37.9%) |
| NA | 0/53 (0.0%) | 3/153 (2.0%) |
| **Water availability[c]** | | |
| Not piped | 14/53 (26.4%) | 32/80 (40%) |
| Piped > 0 and <7 days | 21/53 (39.6%) | 28/80 (35.0%) |
| Piped 7 days | 18/53 (34.0%) | 19/80 (23.8%) |
| NA | 0/53 (0.0%) | 1/80 (1.3%) |
| **Community type** | | |
| Urban | 14/53 (26.4%) | 16/84 (19.0%) |
| Intermediate | 21/53 (39.6%) | 29/84 (34.5%) |
| Rural-road | 15/53 (28.3%) | 32/84 (38.1%) |
| Rural-river | 3/53 (5.7%) | 7/84 (8.3%) |
| **Age (children)** | | |
| 1-week | | 53/153 (34.6%) |
| 3-months | | 20/153 (13.1%) |
| 6-months | | 20/153 (13.1%) |
| 18-months | | 60/153 (39.2%) |
| **Sex (children)** | | |
| Male | | 47/84 (56.0%) |
| Female | | 37/84 (44.0%) |
| **Delivery (children)** | | |
| Vaginal | | 65/84 (77.4%) |
| Cesarean section | | 19/84 (22.6%) |

[a]Households with piped sewers or septic tanks
[b]Households with both piped sewers or septic tanks and piped drinking water
[c]Available for samples at pre-natal visit, and 6- and 18-month visits

low, 18.9% as medium, and 7.5% as high animal exposure groups. Among the children, 37.9% had zero animal exposure and 20.9% were classified as low, 20.3% as medium, and 18.9% as high animal exposure (Table 1).

Examining animal feces as a source of AMR exposure, ESBL-producing bacteria were present across several animal species (e.g., dogs, chicken, cows, pigs and ducks), and ESBL *E. coli* was detected in 88% of chicken fecal samples (Supplementary Fig. S2). Many of these isolates were resistant to third-generation cephalosporins (e.g., ceftriaxone, ceftazidime) (Supplementary Fig. S3).

## Number and abundance of clinically relevant ARGs

Non-metric multidimensional scaling (NMDS) ordination indicates differences in the composition of clinically relevant ARGs between mothers and children (Fig. 1A). In addition, the number of clinically relevant ARGs in mothers (mean = 14, range = [2 – 29]), was significantly lower ($p$-value < 0.05) and less variable compared to children, (28.4 [7 – 95]) (Fig. 2A, B). Clinically relevant ARG abundance ranged from 0.004 to 0.36 coverage/Genome Equivalents (GE) in mothers and from 0.007 to 3.13 coverage/GE in children (Fig. 2C, D). In mothers, the number of clinically relevant ARGs trended lower in both the low ($\beta = -5.67$, 95% CI = −12.83,− 0.51) and medium ($\beta = -5.31$, 95% CI = −11.40, 0.78) animal exposure compared to the high animal exposure group (Fig. 2A and Supplementary Table S2). The number of clinically relevant ARGs in mothers with zero animal exposure was more similar to high animal exposure ($\beta = -3.24$, 95% CI = −9.06, 2.57). In children, both the number and abundance of clinically relevant ARGs increased when animal exposure decreased (Fig. 2B, D and Supplementary Table S3). For example, the number of clinically relevant ARGs was 7.92 units higher in the low animal exposure group compared to the high animal exposure group (95% CI = − 0.51, 16.34).

In mothers, the number of unique clinically relevant ARGs was significantly lower in households with CS&PW ($\beta = -4.16$, 95% CI = − 7.53, − 0.78) (Fig. 2A and Supplementary Table S2). Households with greater access to piped water had lower clinically relevant ARG abundance compared to households without piped water (piped 7-day: − 0.04, 95% CI = − 0.09, 0.01) (Fig. 2C and Supplementary Table S2). In children, both the number of ARGs ($\beta = -5.45$, 95% CI = − 11.12, 0.21), and abundance of ARGs ($\beta = -0.10$, 95% CI = − 0.22, 0.03) trended lower in households with CS&PW compared to households with other systems, though were not statistically significant (Fig. 2B, D and Supplementary Table S3).

When stratified by sex, the overall patterns of associations between exposures and ARG outcomes in children were broadly similar between females and males (Supplementary Tables S6 and S7). Both sexes showed increasing numbers and abundances of ARGs with lower animal exposure, as well as trended towards lower ARG abundance in CS&PW households. However, there were some notable differences: female children in households with CS&PW tended to have higher numbers of unique ARGs compared with other systems ($\beta = 5.96$, 95% CI = 1.41, 10.50]), whereas male children in households with CS&PW had lower numbers of unique ARGs ($\beta = -10.25$, 95% CI = − 17.04, − 3.45). These opposite trends suggest potential sex-specific responses, although variability remained high and associations were not significant in most models.

## Taxonomic diversity, *E. coli* and *K. pneumoniae* relative abundances

Like the resistome, the diversity of child gut microbiomes spanned a wide range. Nonpareil diversity was lower in mothers with medium ($\beta = -1.07$, 95% CI = − 1.70, − 0.43), low ($\beta = -1.58$, 95% CI = − 2.23, − 0.95), and no ($\beta = -1.35$, 95% CI = − 1.96, − 0.75) animal exposure compared to those with high exposure (Fig. 3A). There was no significant difference in sequence diversity when comparing mothers in households with CS&PW to those without ($\beta = 0.01$, 95% CI = − 0.45, 0.43). In children, Nonpareil diversity was not associated with animal exposure and there were no significant differences between children in households with CS&PW and those without in the adjusted models (Animal exposure-Medium: $\beta = 0.10$, 95% CI = − 0.33, 0.53; CS&PW: $\beta = -0.07$, 95% CI = − 0.33, 0.19) (Fig. 3B). Children with 7 days of piped water had significantly lower Nonpareil diversity compared to no piped water ($\beta = -0.63$, 95% CI = − 1.11, − 0.15).

The relative abundances of *E. coli* and *K. pneumoniae* in mothers (*EC*: mean: 0.5% (range: 0-19.8%); *KP*: 0.2 (0–3.8) %) were lower than in children (*EC*: 5.4 (0–53.4) %; *KP*: 2.9 (0–43.6) %) (Fig. 1B). One week old

**A.**

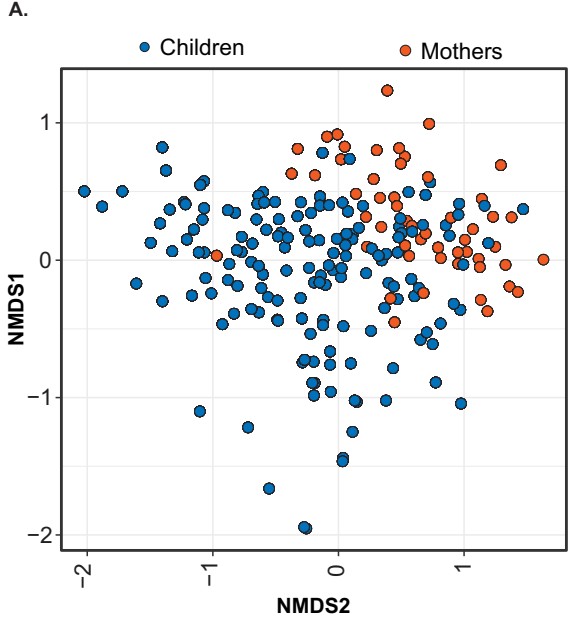

**B.**

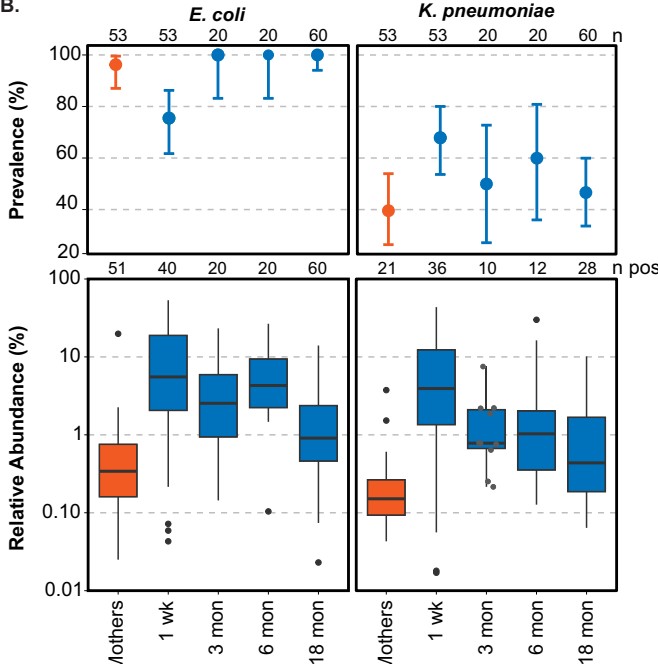

**Fig. 1 | Comparison of resistome and occurrence of *E. coli* and *K. pneumoniae* in mothers and children. A** NMDS ordination of the bray-curtis dissimilarity of clinically relevant ARGs (underlying data in units of coverage/GE). Each point represents an individual sample, with blue points corresponding to children and orange points to mothers. **B** Prevalence and relative abundance of *E. coli* (left) and *K. pneumoniae* (right) in mothers (orange) and children (blue) across age groups. The upper panels show the prevalence (percentage of positive samples) of each species, with error bars representing 95% confidence intervals. The lower panels show boxplots of relative abundance for positive samples, plotted on a log10 scale. Box plots show interquartile range (25th–75th percentiles) with the median and whiskers extending to 1.5 times the interquartile range. Sample sizes (n) are indicated at the top of the prevalence plots for each age group, and the number of positive samples (n pos) is shown below. Individual data points are shown in gray for n ≤ 10.

children had the lowest prevalence of *E. coli* (75.5%) but highest concentration in positive samples (Fig. 1B). *E. coli* relative abundance in the maternal samples with lower animal exposure was not different from those with high animal exposure (Fig. 3C). Similar to ARGs in children, *E. coli* abundance was higher in the zero-animal exposure compared to higher animal exposure group in the unadjusted model ($\beta = 3.06$, 95% CI = −0.01, 6.13) although not significant in the adjusted model ($\beta = 1.05$, 95% CI = −2.34, 4.44) (Fig. 3D). *E. coli* abundance in children from households with CS&PW was lower than those with other systems ($\beta = -3.21$, 95% CI = −6.55, −0.12) (Fig. 3D). *K. pneumoniae* relative abundance was highest in mothers with zero animal exposure compared to other exposure groups and not statistically different from mothers with high animal exposure ($\beta = 0.20$, 95% CI = −0.47, 0.86) (Fig. 3E). *K. pneumoniae* relative abundance was higher in children compared to mothers (Fig. 1B) and showed no consistent trends across exposure categories (Fig. 3F).

When stratified by sex, patterns of taxonomic diversity, *E. coli*, and *K. pneumoniae* abundance in children were broadly similar between females and males (Supplementary Tables S6 and S7). Both sexes showed little association between taxonomic diversity and animal exposure or CS&PW, and *K. pneumoniae* abundances were variable without clear trends across exposures. However, some differences emerged. Female children with continuous piped water (7 days) had significantly lower taxonomic diversity ($\beta = -0.84$, 95% CI = −1.37, −0.30) compared to non-piped water, while there was no difference in male children under the same conditions ($\beta = -0.31$, 95% CI = −0.79, 0.17). Overall, although a few sex-specific differences were observed, most of the exposure outcome relationships showed similar trends in both sexes. Importantly, variability remained high across all models, underscoring the heterogeneity of microbial and resistome outcomes in early childhood.

## Differential associations between animal exposure by WASH access

The associations between animal exposure and the microbial outcomes in mothers follow similar trends between households with and without CS&PW (Fig. 4A). However, some correlations were stronger than others. For example, the number of clinically relevant ARGs was lower in households with low and medium animal exposure, compared to households with high animal exposure, in both groups (households with and without CS&PW) (Supplementary Table S4). However, associations were significant at medium animal exposure in households without CS&PW (no CS&PW/animal expo-medium: $\beta = -11.49$, 95% CI = −21.77, −1.21) and at low animal exposure in households with CS&PW (CS&PW/animal expo-low: $\beta = -8.92$, 95% CI = −16.43, −1.42). The number of clinically relevant ARGs, sequence diversity, and relative abundance of *E. coli* and *K. pneumoniae* were higher in children with less animal exposure in households with CS&PW but not in households without CS&PW (i.e., "Other" systems) (Supplementary Table S5).

## Distribution of clinically relevant ARGs and their ESKAPEE pathogen hosts

Genes that indicate resistance to beta-lactams, specifically $bla_{CTX-M}$, $bla_{OXA}$, $bla_{SHV}$, and $bla_{TEM}$, were found in contigs from all age groups: 37 weeks of pregnancy ($n = 6/53$), one-week-old ($n = 21/53$), three-months-old ($n = 6/20$), six-months-old ($n = 5/20$), and 18-months-old ($n = 29/60$). ESBL genes were consistently found in mothers and children across age groups, although the majority of ARG containing contigs were identified in children overall compared to mothers (Fig. 5A). There was no association between clinically relevant ARG-containing contigs and animal exposure (low exposure in mothers: $\beta = -0.80$, 95% CI = −2.17, 0.60, *p*-value = 0.25; low exposure in

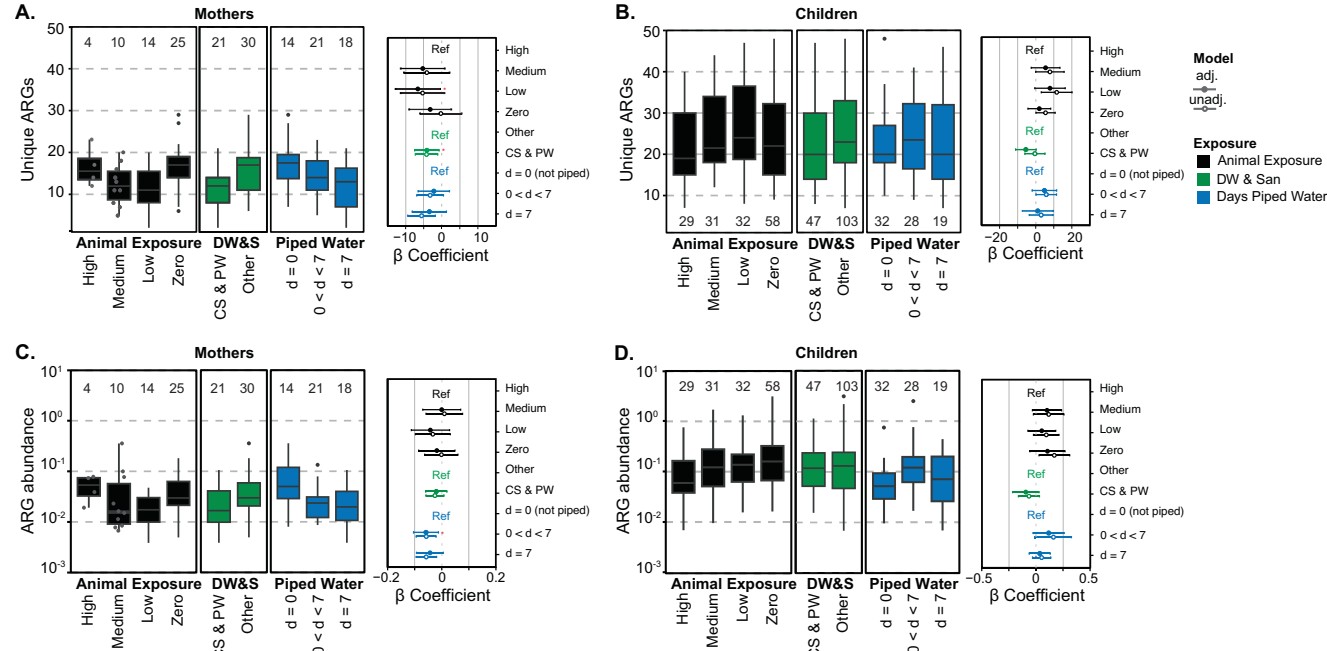

**Fig. 2 | ARG diversity and relative abundance in mothers and children and association with environmental exposures.** ARG diversity (number of unique clinically relevant ARGs) in mothers (**A**) and children (**B**), and normalized relative abundance of ARGs in mothers (**C**) and children (**D**), grouped by animal exposure levels (black), drinking water and sanitation (DW & San, green), and days of piped water availability (blue). The β coefficient for these variables (adjusted-close circle and unadjusted-open circle) are shown to the right of each panel with their 95% CI. Statistically significant β coefficients (*p*-value < 0.05) are marked with a red asterisk in adjusted linear models only. *P*-values are available in Supplementary

Tables 2 and 3. Animal exposure levels are zero exposure (Zero), low exposure (Low), medium exposure (Medium), and high exposure (High; reference in the linear model). Drinking water and sanitation (DW&S) categories were contained sewage and piped water (CS&PW) and other systems (other; reference in the linear model). Piped water availability categories are water available 7 days a week (d = 7), water available less than 7 days a week (0 < d < 7), and not piped systems (d = 0; reference in the linear model). Box plots show interquartile range (25th - 75th percentiles) with the median and whiskers extending to 1.5 times the interquartile range. Individual data points are shown in gray for *n* ≤ 10.

children: $\beta = 1.26$, 95% CI = $-0.28, 2.80$, *p*-value = 0.11). Similarly, the ESKAPEE MAGs were found across age groups and were not associated with animal exposure status (low exposure in mothers: $\beta = 0$, 95% CI = $-0.006, 0.007$, *p*-value = 0.88; low exposure in children: $\beta = 0$, 95% CI = $-0.03, 0.03$, *p*-value = 0.92). *E. coli* and *K. pneumoniae* had the highest coverage and were present in most samples, whereas the coverage of *A. baumannii* was zero in all mother samples (Fig. 5C). We also observed that the coverage estimated from MAGs (coverage in TAD80/GE) and relative abundance from StrainGE were highly correlated for all ESKAPEE pathogens, including *E. coli* ($R = 0.8$, $p < 0.05$) and *K. pneumoniae* ($R = 0.84$, $p < 0.05$) (Supplementary Fig. S4). Both approaches identified *E. coli* and *K. pneumoniae* as the most abundant and frequently detected pathogens across mothers and children. In mothers and children, the ESKAPEE pathogen MAGs harbored diverse clinically relevant ARGs conferring resistance to multiple antibiotic classes. Key beta-lactam resistance genes include $bla_{CTX-M-15}$, $bla_{SHV-106}$, $bla_{TEM-1}$, and $bla_{OXA-120}$, which confer resistance to penicillins, cephalosporins, and carbapenems, and were found in the *K. pneumoniae*, and *A. baumannii* MAGs. Aminoglycoside resistance genes, such as $aac(6')$-$I$ and $aph(6)$-$I$, confer resistance to drugs like gentamicin and kanamycin and were found in *K. pneumoniae* and *E. faecium*. A gene for fluoroquinolone resistance ($qnrB1$) was found in *K. pneumoniae*, while a gene conferring macrolide resistance ($ermB$) was present in the *E. coli* MAG (Fig. 5C).

Plasmid-borne ARGs, were more common than chromosomal ARGs in our dataset; we found 48 ARGs on plasmids and 40 on chromosomes (Fig. 5A). This was particularly true for the genes $bla_{CTX-M-15}$, $bla_{OXA-471}$, and $bla_{SHV-40}$. In contrast, ARGs such as $bla_{SHV-1}$ and $bla_{OXA-1}$ were primarily located on chromosomes. In general, the $bla_{CTX-M}$, $bla_{OXA}$, $bla_{SHV}$, and $bla_{TEM}$ genes are widely distributed across both chromosomal and plasmid DNAs. Consistent with the association of

ESKAPEE pathogen MAGs with beta-lactam resistance, $bla_{CTX-M}$ and $bla_{TEM}$ genes were mainly identified in *E. coli* and *K. pneumoniae*, while $bla_{OXA}$ and $bla_{SHV}$ were typically found in *K. pneumoniae*. Only a few $bla_{CTX-M}$, $bla_{OXA}$, and $bla_{TEM}$ genes were detected in other bacteria, including *Candidatus* Campylobacter infans and *Salmonella enterica* (Fig. 5B).

## Discussion

In this study, we examined how animal exposure and household water and sanitation conditions influence the gut resistome and microbiome in mothers and children. We found stronger associations between animal exposure and resistance patterns in mothers compared to children, and improved water and sanitation were generally linked to lower ARG abundance. Notably, we detected clinically relevant β-lactamase genes, underscoring the public health significance of these findings and the potential role of household exposures in shaping the human resistome.

We found evidence that exposure to animals may be associated with an increased number of clinically relevant ARGs and taxonomic diversity in adults (mothers), while zero exposure appeared more similar to high exposure, likely reflecting indirect exposure routes in these communities. In children, however, no clear or statistically significant associations with animal exposure were observed. Our findings align with existing research indicating that interactions at the human-animal interface can significantly influence the human microbiome and resistome[21,24,46]. While we did not sequence animal fecal microbiomes in this study, ESBL-producing bacteria were prevalent across several animal species. Many *E. coli* isolates were resistant to third-generation cephalosporins (e.g., ceftriaxone, ceftazidime), overlapping with ARG classes identified in the human gut resistome. Although we cannot directly demonstrate transmission between

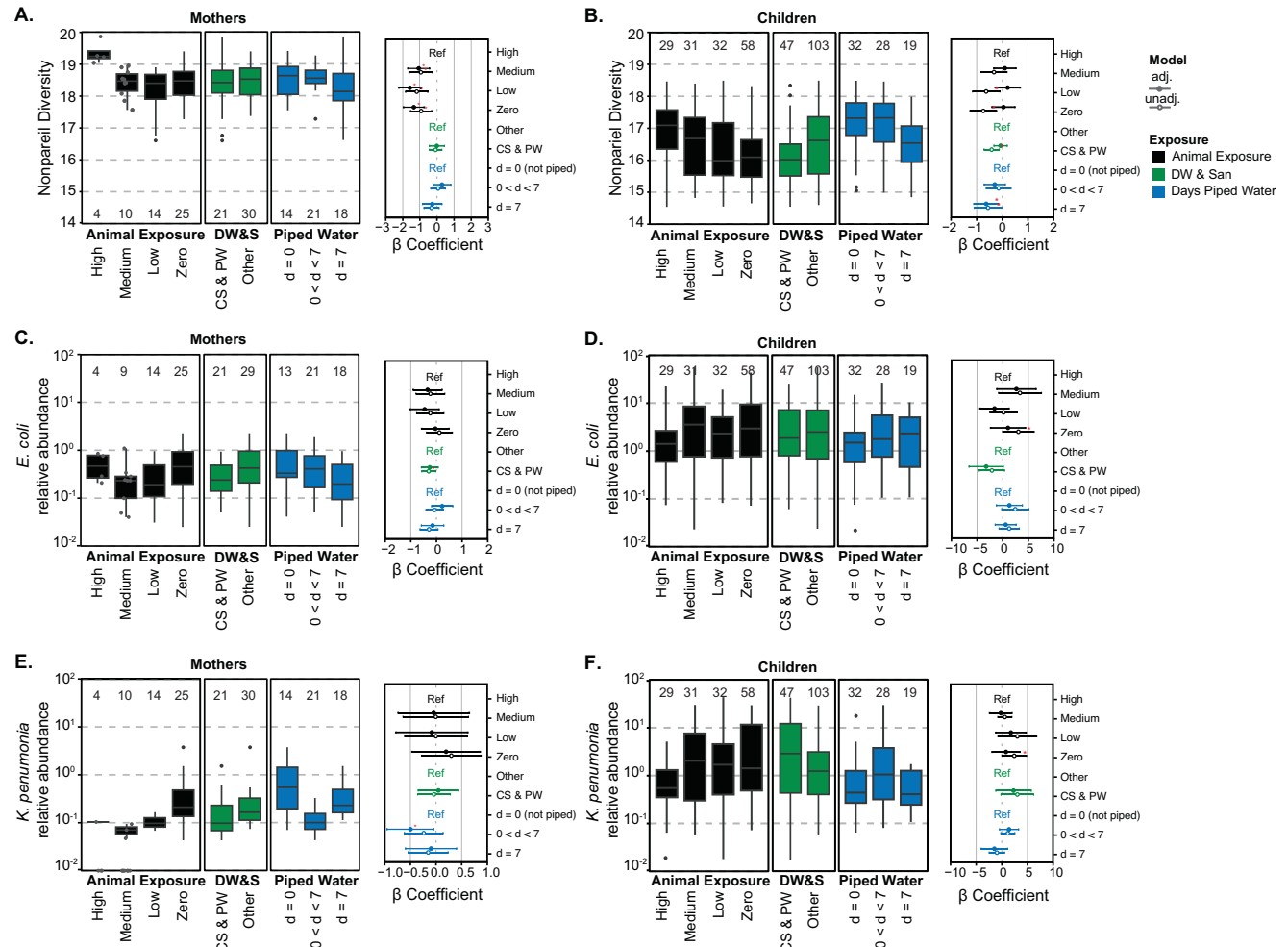

**Fig. 3 | Sequence diversity and relative abundance of *E. coli* and *K. pneumoniae* in mothers and children and association with environmental exposures.** Nonpareil sequence diversity and relative abundance of *E. coli* and *K. pneumoniae* in mothers (**A**, **C**, **E**) and children (**B**, **D**, **F**), grouped by animal exposure levels (black), drinking water and sanitation (DW&S, green), and days of piped water availability (blue). The β coefficients for these variables (adjusted-close circle and unadjusted-open circle) are shown on the right side of each panel with their 95% CI. Statistically significant β coefficients (*p*-value < 0.05) are marked with a red asterisk for adjusted linear models only. *P*-values are available in Supplementary Tables 2 and 3. Animal exposure levels are zero exposure (Zero), low exposure (Low), medium exposure

(Medium), and high exposure (High; reference in the linear model). Drinking water and sanitation (DW&S) categories are contained sewage and piped water (CS&PW) and other systems (Others; reference in the linear model). Piped water availability categories are water available 7 days a week (d = 7), water available less than 7 days (0 < d < 7), and not piped systems (d = 0; reference in the linear model). One mother sample was removed for the relative abundance of *E. coli* since it was greater than two standard deviations away from the mean. Box plots show interquartile range (25th–75th percentiles) with the median and whiskers extending to 1.5 times the interquartile range. Individual data points are shown in gray for *n* ≤ 10.

animals and humans in this study, these results support the plausibility that animals act as reservoirs of clinically relevant ARGs in these communities. We did not find differences in the relative abundance of *E. coli* or *K. pneumoniae* between mothers with low and high animal exposure, and mothers with zero animal exposure trended towards higher relative abundances of *K. pneumoniae* compared to exposed mothers. This aligns with evidence that, although *K. pneumoniae* is common in both humans and animals, direct zoonotic transmission appears likely limited[47]. Our findings differ from studies linking animal contact to increased *E. coli* colonization in humans[48] and from those showing identical *E. coli* strains in pets and owners[49]. This discrepancy may be due to our relatively small high-exposure group, which limited our ability to detect subtle changes in *E. coli* abundance.

Previous studies have demonstrated that livestock[10] and pets[11,12] harbor a wide array of ARGs, which can be transferred to humans through food, direct contact with feces, or environmental pathways[50,51]. Environmental dissemination can occur when bacteria and antibiotic residues from animal production spread through

manure, impacting environmental bacterial populations[51]. Consequently, the environment can become reservoirs of resistance, further facilitating the reintroduction of antimicrobial resistance into human and animal reservoirs[52,53]. The lack of difference between "zero" and high animal exposure levels in mothers in this study suggests that indirect exposure routes (e.g., from environmental fecal contamination) may play an important role in disseminating ARGs. Defining "zero exposure" is particularly challenging in settings where animals are prevalent within the community, as environmental reservoirs (e.g., soil and water bodies) can serve as pathways for ARG transmission via exposure to animal waste, independently of direct animal contact[54]. This complicates the task of accurately defining exposure levels[55].

Interestingly, we did not observe similar associations between animal exposures and ARG diversity and abundance or ESKAPEE pathogen carriage in children. The child resistome and microbiome were more variable compared with mothers. The abundance of ARGs, *E. coli*, and *K. pneumoniae* trended higher in the zero and lower animal exposure groups compared to the high animal exposure group,

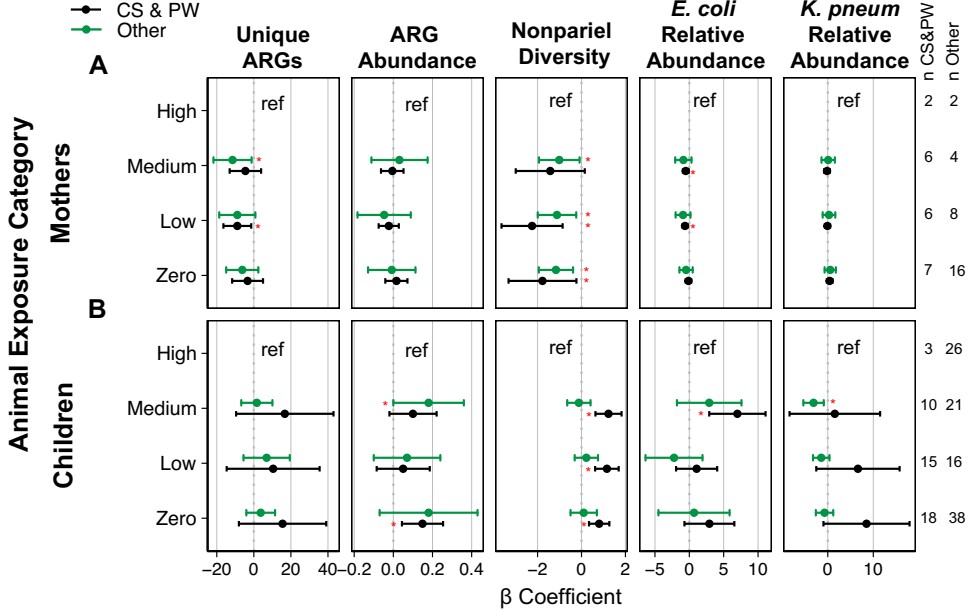

**Fig. 4 | Associations between animal exposure and microbiological outcomes stratified by drinking water and sanitation (DW&S). A** mothers and **B** children. Each plot presents β coefficients (with 95% confidence intervals) between animal exposure categories (High, Medium, Low, and Zero exposure) and the outcomes: Number of clinically relevant ARGs (Unique ARGs), ARGs abundance (coverage/GE), nonpareil sequence diversity, and the relative abundance (%) of *E. coli* and *K. pneumoniae* in households with contained sewage and piped water (CS&PW-black) and without CS&PW (Other-green). Statistically significant β coefficients (*p*-value < 0.05) in linear models are marked with a red asterisk. *P*-values are available in Supplementary Tables S4 and S5.

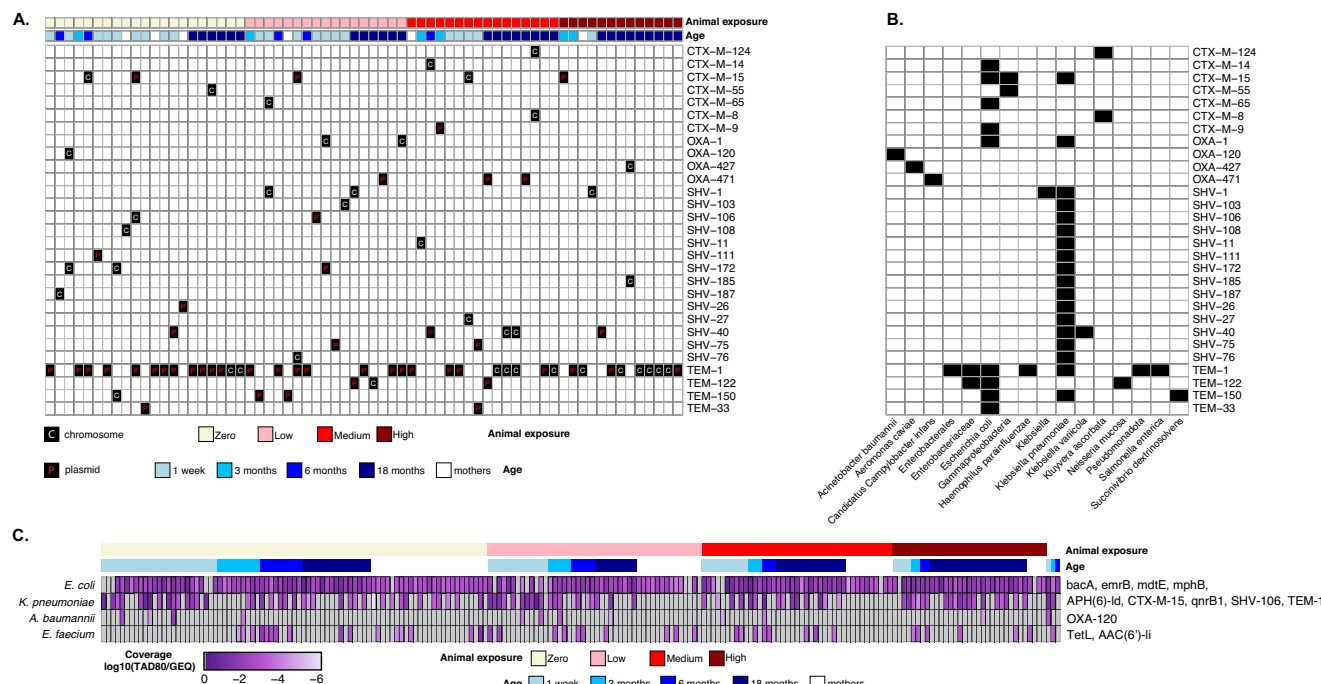

**Fig. 5 | Clinically relevant ARGs and their association with animal exposure, genomic location (chromosomes or plasmids), and linkage to ESKAPEE pathogens. A** Heatmap illustrating the presence of beta-lactamase genes (*bla*CTX-M, *bla*OXA, *bla*SHV, and *bla*TEM, families) on contigs assembled from metagenome sequences across samples. Rows represent specific ARGs assembled in contigs, and columns represent individual samples, categorized by age [mothers (white) and children (shades of blue) at 1 week, 3 months, 6 months, and 18 months] and organized by animal exposure levels [Zero (light yellow), Low (pink), Medium (red), and High (dark red)]. ARGs are classified as chromosomal ("C") or plasmid-borne

("P"). **B** Taxonomic classifications of contigs carrying beta-lactamase genes (*bla*CTX-M, *bla*OXA, *bla*SHV, and *bla*TEM, families) across samples. Rows correspond to specific ARGs, and columns indicate bacterial taxa identified at the highest resolution by Kraken2. Black squares indicate the presence of ARGs within each taxon. **C** Heatmap of MAGs identified as ESKAPEE pathogens containing clinically relevant ARGs. Abundances (log10-transformed coverage) are shown across samples stratified by age and organized by level of animal exposure. Rows represent the MAG (*E. coli, K. pneumoniae, A. baumannii, and E. faecium*), and columns represent individual samples.

though not statistically significant. Several factors may explain differences in associations between mothers and children. First, defining "zero exposure" in households near animals is challenging, which could obscure trends. Second, the dynamic nature of the infant gut microbiome, influenced by breastfeeding, dietary shifts, and infections, likely plays a central role[56,57]. For example, breastfeeding promotes the growth of beneficial bacteria like *Bifidobacterium*[58], while introducing solid foods leads to changes in the composition of the gut microbiome[58,59]. Third, behavioral and environmental differences may result in different types of animal contact for infants compared to mothers. For instance, mothers are more likely to handle animal husbandry tasks such as feeding, cleaning, or managing waste, which increases the chance of direct exposure to animal-associated microbes and ARGs[60]. Infants may be intentionally shielded from animals by caregivers to reduce perceived health risks and direct contact opportunities[61]. Finally, differences in antibiotic exposure and recovery may also explain the divergence between mothers and children. Adult gut microbiomes typically recover from a single antibiotic exposure within about two weeks, whereas children often exhibit less stability, with recovery sometimes extending beyond one month[62]. Such reduced resilience in early life may contribute to the distinct resistome patterns observed in children compared to mothers. These differences likely contribute to the weaker associations observed between animal presence and resistome outcomes in children compared with mothers.

Improved sanitation and water infrastructure (i.e., CS&PW) trended towards lower diversity and abundance of clinically relevant ARGs and a lower relative abundance of *E. coli* in mothers and children, highlighting the role that water and sanitation services play in mitigating microbial transmission and reducing the prevalence of ARGs in the gut. These findings are consistent with previous research showing that access to improved water and sanitation correlated with lower ARG abundance in human gut metagenomes aggregated from 26 countries[63]. While we do not report data on antibiotic consumption or directly assess HGT, prior work has shown that improved WASH conditions can reduce the incidence of infections, which may in turn lower antibiotic use and limit opportunities for ARG dissemination. For instance, in a randomized controlled trial of WASH interventions in Kenya and Bangladesh, WASH interventions were associated with 10-14% lower antibiotic consumption in children in Bangladesh[64]. Improving water and sanitation facilities could decrease the prevalence of ARGs in the environment, reducing human and animal exposure.

Mothers in households with more days of available piped water had lower clinically relevant ARG abundance compared to households without piped water. In regions where piped water supply is unreliable, residents often resort to alternative water sources, such as untreated surface water or storage systems, to meet their daily needs[65,66]. Secondary storage containers typically reduce drinking water quality through recontamination[67]. In addition, intermittent water supply (IWS), characterized by the delivery of water for limited hours or days, can significantly impact water quality through contamination in the pipelines[68,69]. IWS allows for stagnation and depressurization, creating conditions favorable for biofilm growth on pipe surfaces as well as intrusion of environmental bacteria[70]. While studies specifically investigating the impact of IWS on the microbial communities in the human gut microbial are limited, existing research indicates that chlorination to improve drinking water quality has minor impacts on the resistome and microbiome[71].

While previous research indicates that improved WASH infrastructure can reduce microbial transmission and the spread of ARGs[63], we did not find evidence that adequate drinking water and sanitation infrastructure mitigate the impact of animal exposure on the maternal gut microbiome. Conversely, in households with CS&PW systems, children with lower animal exposure had a higher number of clinically

relevant ARGs and taxonomic diversity as well as a higher relative abundance of *E. coli* and *K. pneumoniae*. This counterintuitive finding may indicate that there are alternative environmental contamination pathways contributing to ARG occurrence in children besides WASH and animal exposure. Beyond improving WASH infrastructure, additional measures are likely necessary to address environmental exposures affecting children's gut microbiomes. This may include interventions targeting behavioral practices and broader community-level factors to reduce the burden of antimicrobial resistance[72]. Our findings in the stratified animal exposure analysis may be partially attributed to the limited number of samples with high animal exposure in these settings, which affects the statistical power of our analysis. Therefore, the observed patterns should be interpreted cautiously, and further studies with larger sample sizes are necessary to validate these findings.

The detection of clinically relevant beta-lactamase genes, including $bla_{\text{CTX-M}}$, $bla_{\text{OXA}}$, $bla_{\text{SHV}}$, and $bla_{\text{TEM}}$, across all age groups highlights the presence and persistence of these resistance determinants in the study population. This finding aligns with previous research indicating that certain beta-lactamase genes are prevalent across populations[73]. The higher prevalence of ARGs in younger children compared to mothers aligns with existing research[74,75], even when infants have not been exposed to antibiotics[76]. The high levels of *Gammaproteobacteria*, which are common early gut colonizers and often carry resistance genes, may explain this increased ARG load in the child gut, though the exact source of these resistant strains (whether from the environment, other individuals, or the mother) remains unclear[76].

In our dataset, many clinically relevant ARGs detected in ESKAPEE MAGs were carried on plasmids, including $bla_{\text{CTX-M-15}}$, $bla_{\text{OXA-471}}$, and $bla_{\text{SHV-40}}$, whereas others, such as $bla_{\text{SHV-1}}$ and $bla_{\text{OXA-1}}$ were primarily chromosomal. The presence of plasmid-borne ARGs is consistent with the well-documented role of mobile genetic elements in facilitating the dissemination of resistance traits across bacterial populations. Plasmids have been widely recognized as vectors of ESBL genes among *Enterobacteriaceae*, contributing to the global spread of multidrug resistance[77,78]. While our findings highlight that plasmid-associated ARGs are present within clinically important pathogens in this community, we did not directly assess plasmid mobility, recent HGT events, or de novo ARG acquisition. Additional long-read sequencing and functional assays would be required to determine the dynamics of ARG exchange between bacterial hosts.

*E. coli* and *K. pneumoniae* were the most frequently detected ESKAPEE pathogens in the metagenomes and showed the highest relative abundances across samples. MAGs and StrainGE-identified strains of these species were found in most mother and child samples. In contrast, *A. baumannii* MAGs were notably absent in all mother samples. The observed differences in prevalence and relative abundance of ESKAPEE pathogens may reflect distinct environmental reservoirs or differences in host susceptibility among age groups. *E. coli* is a common inhabitant of the human gut microbiota[79]. Similarly, *K. pneumoniae* is known to colonize the gastrointestinal tract, with prevalence rates varying geographically[80]. In contrast, *A. baumannii* is primarily recognized as an opportunistic pathogen associated with healthcare settings[81]. Among the ESKAPEE MAGs, $bla_{\text{CTX-M}}$ and $bla_{\text{TEM}}$ genes were mainly identified in *E. coli* and *K. pneumoniae*, while $bla_{\text{OXA}}$ and $bla_{\text{SHV}}$ were typically found in *K. pneumoniae*. This observation aligns with previous reports highlighting the association of these genes with ESBL production in these pathogens[82]. Detecting these genes in pathogens of significant clinical concern highlights the importance of continuous surveillance and targeted interventions to mitigate the spread of resistance.

Applying StrainGE for targeted strain detection and assembling MAGs for ESKAPEE pathogens allowed us to directly compare these approaches. StrainGE is a metagenomic tool built to identify and

characterize low-abundance strains within complex microbial communities using short-read metagenomic datasets. It has primarily been used for identifying *E. coli* strains[83]. In our study, StrainGE was specifically used to identify and quantify the relative abundance of other ESKAPEE pathogens in addition to *E. coli*. The relative abundance and presence of ESKAPEE pathogens, as well as overall trends in animal exposure, were consistent between StrainGE and assembled MAGs. Both methods consistently detected the presence of *E. coli*, *K. pneumoniae*, *E. faecium*, and *A. baumannii* across samples, with *E. coli* and *K. pneumoniae* present in more samples and at greater relative abundances. These findings suggest that despite methodological differences, StrainGE and MAG-based approaches yield comparable insights into the prevalence and dominance of clinically relevant pathogens in the gut microbiome. The consistent results obtained from these two approaches support StrainGE as a powerful tool for characterizing other bacterial species beyond *E. coli*. We note, however, that the effectiveness of StrainGE is closely linked to the quality and comprehensiveness of its reference genome database.

We applied a well-considered animal exposure assessment by calculating a score using the framework developed for the FECEZ index[84]. However, accurately categorizing animal exposure levels can be particularly challenging in communities where animals are common, and exposure can occur in multiple settings (additional animal details are available in the supplementary information). This may result in misclassification, which could impact the associations between animal exposure and microbiome or resistome profiles. In our study, the animal exposure score encompassed a range of animals commonly found in Ecuadorian households, including companion animals such as dogs and cats, as well as livestock like poultry, pigs, and cows. Companion animals and livestock differ in their antibiotic exposure and potential to contribute to AMR transmission. Pets often receive antibiotics for medical treatments, and close contact with humans (e.g., petting, licking, and sharing living spaces) facilitates the bidirectional transmission of resistant bacteria[85]. In contrast, livestock are frequently administered antibiotics not only for therapeutic purposes but also for growth promotion and disease prevention[86]. This widespread use contributes to the selection of resistant bacteria within these animal populations, which can be transmitted to humans through direct contact, consumption of contaminated products, or environmental pathways. The inclusion of both companion animals and livestock in our exposure assessment reflects the diverse animal-human interactions present in the study communities. However, the varying degrees of antibiotic exposure and human contact among these animal types underscore the complexity of AMR transmission dynamics.

In addition, the lack of direct sampling from animals prevents a comprehensive investigation of transmission pathways between animals and humans. While evidence suggests resistant bacteria transmission occurs between household animals and humans in this region[25], we were not able to investigate direct AMR transmission without animal microbiome and resistome data. Moreover, our study relied solely on metagenomic short-read sequencing, which limits the ability to directly link ARGs with their bacterial hosts or confirm phenotypic resistance. Culture-based approaches, particularly for *E. coli*, *K. pneumoniae*, and other readily culturable gut pathogens, would provide important complementary data by enabling genomic and phenotypic characterization of isolates and exploring HGT potential. Future studies incorporating direct sampling of both pets and livestock, detailed antibiotic usage data, and complementary isolate-based approaches will enhance our understanding of the pathways and mechanisms of antimicrobial resistance dissemination. In addition, the relatively small sample size, particularly in the high animal exposure groups, limits the statistical power of our study. Finally, although our sex-stratified models suggest similar overall exposure outcome relationships in males and females, a few opposite trends were observed. These differences, while based on limited sample sizes, warrant further investigation into potential biological or behavioral mechanisms underlying sex-specific microbiome and resistome development.

## Methods

### Ethics

The study protocol was approved by the institutional review boards of the University of Washington (UW; STUDY00014270), Emory University (IRB00101202), the University of California, San Francisco (21–33932), the Universidad San Francisco de Quito (USFQ; 2018–022 M), and the Ecuadorian Ministry of Health (MSPCURI000253–4). Written informed consent was obtained for all mothers, both for their own participation and on behalf of their children. Mothers were provided with general health advice, soap, baby oil and small toys (valuing approximately US$3-5) every 3 months.

### Sample and data collection

ECoMiD is a community-based birth cohort study in Ecuador designed to investigate the gut microbiome's interactions with enteric infections and environmental conditions[44]. Team members enrolled women in late pregnancy (~37-weeks) and collected stool samples from mothers and their children starting at one-week post-birth and every three months thereafter, through 24 months of age. The ECoMiD study commenced enrollment in May 2019, with maternal samples collected within one week of consent. Following the initial enrollment phase, which continued through March 2020, the study experienced a pause due to the COVID-19 pandemic. Enrollment resumed in November 2020, and the final child samples were collected in December 2024. ECoMiD study design, sample collection and processing, and other details of data collection have been described previously[44,45]. Briefly, stool samples were collected by caregivers in sterile containers. The samples were aliquoted by field staff and transported in a portable liquid nitrogen tank to the Universidad San Francisco de Quito (USFQ), where they were stored at −80 °C. DNA extraction was performed using Qiagen's QIAamp Fast DNA Stool Mini Kit with modifications described elsewhere[45]. Frozen DNA samples were cold-shipped from USFQ to the University of Washington (UW), Georgia Institute of Technology (Georgia Tech), and stored at −80 °C until metagenomic sequencing.

Environmental and demographic variables were collected through structured questionnaires and fieldworker observations. The sex of children was reported by mothers at the one-week visit. The animal exposure score (calculated based on the FECEZ index[84] - additional animal details in the SI), urbanicity classification, and socioeconomic status (SES) calculations are described in detail elsewhere[45]. Briefly, we developed a score to measure exposure to animals and animal feces in all households enrolled in the ECoMiD study based on survey questions regarding animal ownership, animal-related behaviors, the presence of animal feces, and the presence of animals (score: 0-4)[45,84]. The score was categorized for the subset of samples in this study into none (0.00), low (0.00 − 0.55), medium (0.55 − 0.90), and high (> 0.90) animal exposure, where the low, medium, and high cutoff values were based on tertiles (i.e., Lower tertile, Middle tertile and Upper tertile) calculated using the samples considered in this study. Communities were grouped into four categories with varying urbanicity: (1) Urban: the city of Esmeraldas (population ~162,000), (2) Intermediate: the town of Borbón, (population ~5000), (3) Rural-road: rural villages that are accessible by road near Borbón, (populations ~500–1000), and (4) Rural-river: rural villages that are mostly accessible by river with some limited car accessibility, (populations ~200–700). An asset score was created as a proxy for SES using ownership data on 15 household possessions and multiple correspondence analysis. The asset score was divided into quartiles, representing SES, with 1 indicating the lowest and 4 the highest SES[87].

Survey data on household environmental exposures were collected from mothers at ~37-weeks of pregnancy and from children at

6- and 18-months of age. Consequently, animal exposure data for children were obtained from the nearest available timepoints: one-week data from the prenatal visit and three-month data from the 6-month visit. Drinking water, sanitation, and covariates in the linear models (e.g., community type, SES) were derived from survey responses at the pre-natal visit, as this information showed minimal variation over the course of the study. However, because piped water availability (days per week) was recorded for the week prior to each visit and this metric often varied over time, only samples from visits with this information (pre-natal visit, and 6- and 18-month visits) were included in the analysis.

### Animal feces culture and isolation

For the purposes of verifying animal exposure as a source of antimicrobial-resistant bacteria in our study setting, we collected ESBL-producing bacterial isolates from animals in a subset of the same households. As part of the ECoMiD AnEx sub-study, 47 fecal samples were obtained from dogs, chickens, pigs, cows, and ducks and cultured to screen for ESBL-producing bacteria. Presumptive *E. coli* isolates were subsequently subjected to antimicrobial susceptibility testing following CLSI guidelines[88]. Further methodological details are provided in the Supplementary Information. We note that animal fecal samples were not included in the metagenomic analysis.

### Metagenomic sequencing and assembly

We performed shotgun metagenomic sequencing on 206 stool samples from 84 households enrolled in the ECoMiD study. Samples were collected from pregnant women at ~37-week gestation ($n = 53$) and their children at 1-week ($n = 53$), 3-months ($n = 20$), 6-months ($n = 20$), and 18-months-old ($n = 60$). Our study included 53 mother–infant pairs, with infants sampled at one week of age. The exact selection of samples is presented in Supplementary Table S1. A total of 53 samples from mothers and 93 samples from children (1-week, 3-months and 6-months) were sequenced at the Georgia Institute of Technology (Georgia Tech) Sequencing Core facility. At Georgia Tech, DNA extracts were subject to the Illumina DNA library preparation kit with 25–50 ng of input and 5 PCR cycles followed by sequencing on an Illumina NovaSeq 6000 instrument using a 2 × 150 bp paired-end kit with 300 cycles. An additional 60 stool samples from 18-month-old children were sequenced at UW. UW DNA extracts were prepared by the Microbial Interactions and Microbiome Center (mim_c) using the Illumina DNA library preparation kit with 10 ng of input and 12 PCR cycles, followed by sequencing on an Illumina NextSeq 2000. Three six-month-old samples were sequenced by both mim_c and Georgia Tech to ensure comparability of results (Supplementary Fig. S1). Raw data for the metagenomes was deposited in NCBI under BioProject PRJNA1225421.

The average shotgun metagenome read count was ~23.8 (range 6-80) million reads per sample. Human reads were removed from the metagenomes using bbtagger v3.101[89] while read trimming was performed using multitrim v1.2.6[90]. We first created a fasta file index for bmfilter and srprism using the commands *bmtool* and *srprism mkindex*, respectively. Then, we used blast v2.5.0[91] to create a blast database using the command *makeblastdb* to remove the human reads with the command "bmtagger.sh". Filtered metagenome reads were assigned taxonomy using kraken2 v2.1.3[92] and diversity was calculated using the Nonpareil diversity index[93,94]. Filtered reads were assembled using metaSpades v3.15.5[95] with kmer sizes 21, 33, 55, 77, 99 and 127. The resulting assemblies were subject to gene calling using Prodigal v2.6.3[96] and ARGs were annotated using the Comprehensive Antibiotic Resistance Database (CARD) v3.2.5[97,98] using abricate v1.0.1[99]. Contigs containing ARGs were taxonomically annotated with kraken2 v2.1.3 and these annotations were corroborated with blast. PlasX v0.0.0 was used for the detection and classification of plasmids within assemblies[100].

### Metagenome assembled genomes (MAGs) recovery, annotation, and dereplication

MAGs were recovered de novo from the assemblies of the first set of metagenomes sequenced at Georgia Tech (see metagenomic sequencing and assembly) after removing contigs less than 5000 bp using MetaBAT2 v2.12.1[101], and MaxBin2 v2.2.7[102]. MAGs were dereplicated using dRep v3.4.0[103] at 95% ANI. MAG quality was evaluated with CheckM v1.2.2[104] and taxonomy was determined with the Genome Taxonomy Database Toolkit (GTDB-Tk v2.3.2, database release 214)[105]. MAGs with a CheckM quality scores ≥50, calculated as "Quality = Completeness – (5 x Contamination)", were selected, resulting in 285 MAGs used in further analyses. Prodigal v2.6.3[96] was used to call genes, and ARGs were annotated against the CARD database[97,98] using ABRi-cate v1.0.1[99].

### Antimicrobial resistance genes (ARGs)

Trimmed metagenome reads were mapped to the CARD database v3.2.5 using the DIAMOND v2.0.15 aligner (--id 95 --query-cover 80 -e 0.00001 -k 1)[106]. We used the *BlastTab.seqdepth.pl* script from the enveomics toolkit to assess sequence depth for genes[107]. Genome equivalent (GE) values were estimated using the MicrobeCensus tool v1.1.1[108]. ARG relative abundance was estimated by normalizing the coverage of clinically relevant ARGs, as reported in Zhang et al. [109], to genome equivalents (coverage/GE or % of total genomes/cells that carry the corresponding gene, assuming 1 gene copy per genome). In addition to the ARGs identified by Zhang et al., we expanded the list of clinically relevant ARGs to include all $bla_{CTX-M}$, $bla_{SHV}$, $bla_{OXA}$, and $bla_{TEM}$ alleles (see Supplementary Data) due to the widespread prevalence and clinical significance of these β-lactamase genes[110]. By encompassing all alleles of these genes, our analysis aims to capture the full spectrum of clinically relevant β-lactamase-mediated resistance, ensuring a comprehensive assessment of ARG diversity and abundance in our metagenomic samples. The alpha diversity of ARGs was defined as the number of unique clinically relevant ARGs (Supplementary Data) per sample.

### ESKAPEE pathogens

The presence and relative abundances of ESKAPEE pathogen strains were determined from the metagenomic data using the StrainGE toolkit v1.3.9[83], which facilitates strain-level analysis and relative abundance quantification from metagenomes. First, a custom reference database was constructed using representative genome sequences of each of the ESKAPEE pathogens. All complete genome sequences for each ESKAPEE pathogen were downloaded from the NCBI RefSeq database[111] to ensure comprehensive strain representation. All references were organized in a single directory using the script "prepare_strainge_db.py". Then, reference sequences were kmerized using the *straingst kmerize* command. Redundant reference genomes were removed by computing the pairwise similarities between k-mer sets using the command *straingst kmersim,* and references were clustered using *straingst cluster*. Because StrainGST compares the k-mer profiles of references in the database to the k-mers in the sample to identify close reference genomes to strains in a sample, we also kmerized the sample metagenome reads using the *straingst kmerize* command. *Straingst run* was used to list the identified reference strains and associated metrics. Concatenated reference fasta files, containing a close reference genome for each strain identified in the sample, were created with *straingr prepare-ref*. Reference fasta files for each strain identified in the sample were indexed with *bwa index*, and sequencing reads from each sample were aligned to the corresponding reference genome using *bwa mem*[112]. Finally, we ran *straingr call* to call all strain variants in the samples and associated data, including strain name, relative abundance, and coverage in the metagenome sequencing data. To evaluate the consistency of ESKAPEE pathogen detection, we compared MAG-based coverage to relative abundances obtained using StrainGE by performing linear regressions.

## Statistical analyses

Statistical analysis was performed using R v4.4.1. We used multivariate linear regression models to determine associations between environmental exposures (animal exposure, sanitation facilities, drinking water source, piped water availability) and taxonomy diversity (Nonpareil sequence diversity), clinically relevant ARGs abundance, number of clinically relevant ARGs (ARGs diversity) and ESKAPEE pathogen relative abundance (i.e., *E. coli* and *K. pneumoniae*). Pair-wise significances were calculated with the Wilcoxon test.

**Statistics & reproducibility.** No statistical method was used to predetermine sample size. The initial 53 mother and 93 child stool samples were selected based on collection prior to the COVID-19 pandemic project pause. A second set of 60 stool samples from 18-month-old children were selected to expand our analysis of environmental exposures. Thirty of these samples correspond to the pre-COVID-19 enrollment period and children who were already included in earlier time points (e.g., 1 week, 3 months, and 6 months). This allowed us to assess exposure over a longer developmental window (18 months). The remaining 30 samples were collected from children enrolled after the COVID-19 pause. These samples were selected to obtain a more balanced distribution of animal exposure across households. No data were excluded from analyses unless specified. The sample collection was not randomized. The Investigators were not blinded to exposures during data analysis. Multivariate linear regression models were replicated by the study Investigators.

**Exposure definitions.** We considered the environmental exposures: (i) animal exposure score, (ii) combined sanitation & drinking water, and (iii) piped water availability for both mothers and children. The animal exposure score was defined using survey questions regarding animal ownership, animal-related behaviors, the presence of animal feces, and the presence of animals, as described above. For combined sanitation and drinking water, we divided the households into two categories: those with access to both contained sewage and piped water (CS&PW) versus "other". Contained sewage was defined as households with sewer systems or septic tanks. Our definition of sanitation includes household services only, meaning that contained sewage did not indicate wastewater treatment. Piped water was defined as piped water reported as a household's drinking water source. The "other" category included households who used latrines or other forms of sanitation, and bottled water, surface water, or rainwater for drinking. Categories were developed to reflect the variation in drinking water and sanitation conditions in our study area and that did not overlap with community type. In addition, we evaluated associations with water availability, measured as the reported number of days households had piped water service during the week prior to sampling. Households were categorized as piped 7 days per week (d = 7), piped less than 7 days (0 < d < 7), and no piped water (d = 0).

**Outcome definitions.** Outcome variables were: (i) clinically relevant ARG abundance, (ii) number of unique clinically relevant ARGs (ARG diversity), (iii) Nonpareil sequence diversity, and (iv) the relative abundance of the ESKAPEE pathogens *E. coli* and *K. pneumoniae*. We created databases and ran strain-identifying analyses for all ESKAPEE pathogens, but only *E. coli* (93%) and *K. pneumoniae* (52%) were detected in more than 20% of metagenomes. *A. baumannii* (3%) and *E. faecium* (11%) were also detected, but with insufficient prevalence. We therefore only included *E. coli* and *K. pneumoniae* in the linear model analyses.

**Multivariate linear regression models.** Multivariate linear regression models were performed separately for mothers and children. We separated adults and children because the gut microbiome of children evolves rapidly during the first years of life, whereas adults typically have a more stable and diverse microbiome that is clearly distinguishable from children[113]. In addition, children interact with their environments differently than mothers. For example, young children crawl, play on the ground, and put objects in their mouths. We used generalized linear models (GLM) with R's glm() function for the maternal samples and controlled for SES and community type. Because child samples were longitudinal and involved repeated measures from the same individuals at different ages, we used generalized estimating equations (GEE) with the geeglm() function in R, accounting for within-child correlation[87]. For the child models, we controlled for SES, community type, age, birth mode, number of people in a household, and exclusive breastfeeding. Planned vaginal delivery was part of the inclusion criteria for the ECoMiD study; however, children delivered by unplanned emergency c-section were not excluded. Collinearity among predictors was evaluated by calculating variance inflation factors (VIF). Any covariates with a VIF greater than 4 were sequentially removed from the models to avoid multicollinearity. As a result, breastfeeding was removed since it was correlated with age.

**Interaction between WASH variables and animal exposure.** We investigated whether drinking water and sanitation conditions modified the relationship between animal exposure and our outcomes of interest. Our effect modifier of interest was a binary variable indicating if the mother or child lived in a household with both contained sewage and piped water (CS&PW) versus a household with other combinations of water and sanitation systems (Other). We utilized the *waldtest* function in R to test for interaction between CS&PW and animal exposure. These interactions were assessed in adjusted models that included the covariates community type and SES for both mothers and children. Age and birth mode were also included in the child models.

**Stratified models by child sex.** To assess whether associations between environmental exposures and gut microbiome or resistome outcomes differed by sex, we conducted stratified analyses for female and male children. All models described above were repeated separately within each sex category. The exposures included animal exposure, drinking water and sanitation conditions, and piped water availability, and the outcomes were the number and abundance of clinically relevant ARGs, taxonomic diversity, and the relative abundances of *E. coli* and *K. pneumoniae*. The same covariates applied in the primary child models (community type, SES, age, number of people in a household, and birth mode) were retained in these analyses. Effect estimates (β coefficients) with 95% confidence intervals were obtained for each stratified model.

### Reporting summary

Further information on research design is available in the Nature Portfolio Reporting Summary linked to this article.

## Data availability

All fastq files generated in this study have been deposited in the Sequence Read Archives under BioProject PRJNA1225421. All MAG assemblies have been deposited under accession numbers SAMN52012661- SAMN52012945. Data used for linear models are provided in the supporting information and on GitHub (https://github.com/fuhr-microlab/ecomid_amr). Source data are provided in this paper.

## Code availability

Analysis codes are available on GitHub (https://github.com/fuhr-microlab/ecomid_amr)[114].

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

## Acknowledgements

We thank the participants of the ECoMiD study and the dedicated field staff who administered the surveys and collected samples. Sequencing support for 60 samples was provided by the University of Washington Microbial Interactions & Microbiome Center (mim_c). I.C. was supported under the ASEE e-fellows program Federal Award 2127509. E.R.F. was supported by the University of Washington Interdisciplinary Center for Exposures, Diseases, Genomics & Environment under grant P30 ES007033. V.A. and the animal fecal antimicrobial susceptibility testing was supported by the UW Department of Global Health - Thomas Francis Jr. Health Fellowship 2023-2024. This work was funded by the National Institutes of Health (R01AI137679 and R01AI162867). The content is solely the responsibility of the authors and does not necessarily represent the official views of the National Institutes of Health.

## Author contributions

Project conceptualization was performed by I.C., K.L. and E.R.F. Samples were collected by V.A. and the ECoMiD Team. I.C., A.D.V, K.J.J., N.A.Z., C.H., A.M.B., G.T., K.T.K., K.L. and E.R.F contributed to methodology. Laboratory work were performed by V.A., C.S.F.S. and J.K.H.

Data analysis was conducted by I.C., V.A., A.D.V., and C.J.R. Visualization of data and results was performed by I.C., V.A. and E.R.F. Funding for this work was acquired by I.C., K.J.J., G.O.L., J.N.S.E., G.T., K.T.K., K.L. and E.R.F. Writing of the original draft was carried out by I.C., V.A., K.L. and E.R.F. Reviewing and editing of the manuscript was performed by all.

## Competing interests

The authors declare no competing interests.

## Additional information

[1]Department of Environmental and Occupational Health Sciences, University of Washington, Seattle, Washington, USA. [2]School of Civil and Environmental Engineering and School of Biological Sciences, Georgia Institute of Technology, Atlanta, Georgia, USA. [3]Department of Population Health Sciences, Georgia State University, Atlanta, Georgia, USA. [4]Rutgers Global Health Institute, Rutgers University, New Brunswick, New Jersey, USA. [5]Department of Biostatistics and Epidemiology, Rutgers School of Public Health, New Brunswick, New Jersey, USA. [6]Emory School of Medicine, Emory University, Atlanta, Georgia, USA. [7]Department of Epidemiology, University of Michigan, Ann Arbor, Michigan, USA. [8]Instituto de Microbiología, Colegio de Ciencias Biologicas y Ambientales, Universidad San Francisco de Quito, Quito, Ecuador. [9]Department of Civil and Environmental Engineering, University of Washington, Seattle, Washington, USA. [15]These authors contributed equally: Viviana Albán, Ana Durán-Viseras. ✉e-mail: klevyx@uw.edu; efuhrm@uw.edu

## the ECoMiD Authorship Group

**Principal Investigators/Co-Investigators** Karen Levy ®[1]✉, Joseph N. S. Eisenberg[7], Gwenyth O. Lee[4,5], Gabriel Trueba ®[8], Benjamin F. Arnold[10], Konstantinos T. Konstantinidis[2] & William Cevallos[11]

**Field Data Collection Subgroup** Adriana Lupero[8], Mauricio Ayoví[8] & Molly K. Miller-Petrie[1]

**Data Management Subgroup** Jesse Contreras[7] & Jessica Uruchima[7]

## Laboratory Analysis Subgroup

**Lab coordination** Christine Fagnant-Sperati[1], Gabriela Vasco[12] & Stuart Torres[8]

**Gut microbiome** Janet Hatt[2], Ana Durán-Viseras ®[2,15] & Kelsey Jesser[1]

## Animal Exposure Subgroup

**Qualitative & Survey Data** April Ballard[3], Bethany Caruso[13] & Betty Corozo[14]

**Microbiology** Kelsey Jesser[1], Viviana Albán ®[1,15], Gabriel Trueba ®[8] & Analía Galarza[8]

[10]University of California San Francisco, Proctor Foundation and Department of Opthamology, San Francisco, CA, USA. [11]Universidad Central del Ecuador, Instituto de Biomedicina, Quito, Ecuador. [12]Universidad Central del Ecuador, Facultad de Ciencias Médicas, Carrera de Medicina, Quito, Ecuador. [13]Emory University, Department of Global Health, Atlanta, Georgia, USA. [14]Universidad Técnica Luis Vargas Torres de Esmeraldas, Esmeraldas, Ecuador.

