## [Transparent Peer Review file · Nature Communications]

Environmental exposures associated with the gut microbiome and resistome of pregnant women and children in Northwest Ecuador

Corresponding Author: Dr Erica Fuhrmeister

Version 0:

Reviewer comments:

Reviewer #1

(Remarks to the Author)

This manuscript reported a metagenomic analysis of the human gut resistome and microbiome in a cohort of 53 pregnant women and 84 children residing in Northwest Ecuador. The authors investigate the influence of environmental exposures—specifically water, sanitation, and hygiene (WASH) infrastructure and animal contact—on the abundance and diversity of antimicrobial resistance genes (ARGs) in human gut microbiota. The study finds that children harbor a higher abundance of ARGs compared to mothers, and that certain environmental conditions, such as improved sanitation and longer duration of piped water access, are associated with reduced ARG diversity and abundance in mothers. The authors also report that high levels of animal exposure are correlated with increased microbiome diversity and a trend toward greater ARG richness. In contrast, environmental associations were less pronounced in children, possibly due to the inherent variability of the early-life gut microbiome.

The study offers valuable insights into the dynamics of antimicrobial resistance in vulnerable populations and provides evidence supporting the potential benefits of improved infrastructure. However, there are concerns regarding the structure and clarity of the analyses, including the rationale for the narrowed focus on certain resistance genes, the framing of household relationships, and the completeness of key contextual information. Addressing these issues would improve the manuscript and enhance its relevance for a broader scientific audience.

Major comments:

- Justification of Focus on β -lactam ARGs: The rationale for focusing exclusively on β -lactam resistance genes should be more clearly justified in the Introduction. I recommend beginning with a broad characterization of the resistome in infants and mothers—identifying the most prevalent ARGs and their associated bacterial hosts—followed by a specific analysis of ARGs in ESKAPE pathogens. This approach would provide a more comprehensive and unbiased overview of the infant resistome and better justify the narrowed focus.
- Birth Mode and Feeding Information: Please report the number of infants delivered vaginally and the number who were exclusively breastfed. Breastfeeding influences the microbiome and the resistome. This information is crucial to contextualize resistome development.
- Mother–Infant Pairs: Clarify the number of complete mother–infant pairs included in the study. In Section 3.1, households should be described based on cohabitating mother–infant pairs rather than treating mothers and infants separately. What is the size of the family? did you check the impact of caregivers, siblings and other members living together?
- Lines 400–411: Please, a comparison between mother and infants will be interesting. Please, include a figure that compares mothers and infants to complement this section.
- Did the authors analysed the pets microbiota to confirm the origin/transfer of ARGs to the families?
- Lines 461–469: It would enhance clarity to display all MAGs and their predicted resistance profiles in a phylogenetic tree.
- Discussion – Opening Summary: Begin the Discussion section with a succinct summary of the study's key findings.
- Line 481–482: This statement appears to contradict the findings described in lines 360–362. Please revise to ensure internal consistency.
- Line 578–580: The current phrasing suggests that the number of recovered MAGs directly reflects the relative abundance of microbes in the microbiome. However, this is misleading, as MAGs recovery is more strongly influenced by sequencing depth and assembly efficiency than by actual microbial abundance. Please revise this statement.
- The comparison of StrainGE results and MAGs is not included in the Results section. Please include or explicitly justify its

omission.

Minor comments

- Consistently use a single scheme for tertile classification—either "high, medium, low" or "upper, middle, lower"—throughout the text and figures
- Panels C and D of Figure 1 and Panel B of Figure 3 are not mentioned in the main text. Please ensure all figure panels are explicitly cited and discussed.
- Line 390: Include a reference to the appropriate panel letter of Figure 2.
- Figure 4 Caption (Line 436): Do not re-abbreviate "antibiotic resistance genes" (ARGs) in the figure caption if the abbreviation has already been defined.
- Line 454: specify the age of the children you are comparing here.
- Indicate the sample size (n) for each group in the relevant figures for better interpretability.
- Line 556: The claim made here is too strong, as it is based solely on b-lactamases without comparison to other ARG classes. Please rephrase to reflect the scope of your data.

Reviewer #2

(Remarks to the Author)

With this study, Cotto et al. explored the relationships between environmental exposures (e.g., animal contact, water and sanitation access) and the human gut resistome by leveraging metagenomic sequencing and statistical modeling in a large cohort of mothers and children in rural Ecuador. Using a combination of shotgun metagenomic sequencing, strain-level profiling, and generalized estimating equations, they assess how animal exposure and WASH (water, sanitation, and hygiene) infrastructure impact gut microbial diversity, ARG (antibiotic resistance gene) abundance and diversity, and the presence of clinically relevant ESKAPEE pathogens. They found that (1) mothers with higher animal exposure showed greater taxonomic diversity and marginally higher ARG diversity; (2) improved WASH infrastructure (specifically combined access to piped water and contained sewage) was associated with significantly lower ARG abundance and richness; and (3) ARGs, including beta-lactamases, were frequently detected in ESKAPEE pathogens such as *E. coli* and *K. pneumoniae*, often on plasmid-borne contigs. This is an important and timely study, and the authors successfully conducted detailed metagenomic work in a challenging field setting, particularly involving sampling of both adults and young children. The integration of environmental context with high-resolution resistome data is a valuable contribution to the growing literature on antimicrobial resistance in low- and middle-income settings. However, despite the study's strengths in large-scale data collection and bioinformatic analysis, several important mechanistic gaps and interpretive assumptions remain unresolved. Below are some specific concerns and suggestions for improvement.

1. The authors suggest that increased ARG diversity in mothers with high animal exposure may result from microbial spillovers from animals. However, this claim remains speculative, as the study does not provide taxonomic resolution of ARG carriers nor track gene flow between animal microbiota and the human gut. No taxonomic enrichment analyses, ARG-host assignment, or co-occurrence networks were performed to support this conclusion. Furthermore, animal fecal microbiomes were not sequenced, limiting source attributions. The authors may consider performing metagenomic co-assembly and genome binning to link ARGs to host taxa, or may use co-occurrence or network-based analysis to identify bacterial lineages enriched in high-exposure groups. In addition, sequencing animal microbiomes and comparing shared ARG profiles, although they have mentioned, need to have animal fecal data.
2. While the reduction in ARG richness and abundance in mothers with CS&PW infrastructure is statistically robust, the mechanistic interpretation that WASH reduces ARGs via lower antibiotic use and fewer HGT events is unsupported by direct evidence. No data on antibiotic consumption was collected, nor were HGT rates or mobile genetic elements explicitly analyzed.
3. The findings for children differed notably from those for mothers, with animal exposure showing no consistent trends and WASH having weaker or nonsignificant associations. This divergence was not deeply discussed or contextualized. The authors should address why children's resistome patterns do not mirror their mothers'. Are children less exposed due to behavioral factors? Does gut microbial development or antibiotic history play a role? Including hypotheses or preliminary comparisons would improve clarity.
4. The detection of clinically relevant ARGs in ESKAPEE MAGs is important, but this study does not demonstrate actual transfer of ARGs into these pathogens during the study period. Many of the ARGs reported (e.g., blaCTX-M, blaTEM) are well known and widely distributed. Without longitudinal sampling, full plasmid resolution (e.g., via long-read sequencing), or direct functional assays, it is not possible to infer recent horizontal gene transfer or de novo ARG acquisition.
5. Moreover, the study would benefit from complementary culture-based validation. Given that *E. coli*, *K. pneumoniae*, and other ESKAPEE pathogens are routinely culturable from stool, the authors could have attempted to isolate these strains and screen them phenotypically or genomically for ARG content. Cultured those isolates would be better for direct correlation of ARGs with host genotype and phenotype. In short, in addition to metagenomic inference, the author should consider strengthening the study with isolate-based approaches — particularly for common gut pathogens that are straightforward to culture and characterize.

Reviewer #3

(Remarks to the Author)

Reviewer #4

(Remarks to the Author)

Version 1:

Reviewer comments:

Reviewer #1

(Remarks to the Author)

Authors have covered my questions and concerns. Paper is improved.

Reviewer #2

(Remarks to the Author)

Thank you for the revisions. The manuscript is clearer, several claims are tempered, and the added culture snapshot is informative. However, there's no substantive evidentiary advance (e.g., taxonomic enrichment, co-occurrence/network analysis, robust ARG–host linkage, antibiotic/MGE data, or human-isolate validation), so our core comments remain not fully addressed.

Reviewer #3

(Remarks to the Author)

Reviewer #4

(Remarks to the Author)

Reviewer #1 (Remarks to the Author):

This manuscript reported a metagenomic analysis of the human gut resistome and microbiome in a cohort of 53 pregnant women and 84 children residing in Northwest Ecuador. The authors investigate the influence of environmental exposures—specifically water, sanitation, and hygiene (WASH) infrastructure and animal contact—on the abundance and diversity of antimicrobial resistance genes (ARGs) in human gut microbiota. The study finds that children harbor a higher abundance of ARGs compared to mothers, and that certain environmental conditions, such as improved sanitation and longer duration of piped water access, are associated with reduced ARG diversity and abundance in mothers. The authors also report that high levels of animal exposure are correlated with increased microbiome diversity and a trend toward greater ARG richness. In contrast, environmental associations were less pronounced in children, possibly due to the inherent variability of the early-life gut microbiome.

The study offers valuable insights into the dynamics of antimicrobial resistance in vulnerable populations and provides evidence supporting the potential benefits of improved infrastructure. However, there are concerns regarding the structure and clarity of the analyses, including the rationale for the narrowed focus on certain resistance genes, the framing of household relationships, and the completeness of key contextual information. Addressing these issues would improve the manuscript and enhance its relevance for a broader scientific audience.

Response: We thank the reviewer for the thoughtful comments. We have addressed the concerns as specified below.

Major comments:

- Justification of Focus on β -lactam ARGs: The rationale for focusing exclusively on β -lactam resistance genes should be more clearly justified in the Introduction. I recommend beginning with a broad characterization of the resistome in infants and mothers—identifying the most prevalent ARGs and their associated bacterial hosts—followed by a specific analysis of ARGs in ESKAPE pathogens. This approach would provide a more comprehensive and unbiased overview of the infant resistome and better justify the narrowed focus.

Response: We thank the reviewer for this comment. To clarify, our study did not analyze only β -lactam resistance genes but instead included a broad assessment of all clinically relevant ARGs in the infant and maternal resistome. Clinically relevant resistance genes were defined based on previous work by Zhang et al.¹ that assessed resistance genes by human-associated, mobility, and presence in pathogen genomes. We added to this list with additional β -lactam resistance genes that are important in the local context. This list is available in the supporting information 2 (Table S2). Within this broader analysis, we placed particular emphasis on β -lactam ARGs because of their clinical significance and the evidence from prior studies in Ecuador demonstrating their widespread occurrence in both humans and animals. To further clarify this, we modified the introduction to justify this focus. Specifically, we added findings from previous research in Ecuador documenting the high prevalence of ESBL-producing Enterobacterales. Finally, we included additional data on the high prevalence of cultured ESBL-producing bacteria

in animal feces in the supporting information. ESBL *E. coli* and *Pseudomonas/Acinetobacter* was present in 100% of bovine samples and most poultry fecal samples (*E. coli*=88% and *Pseudomonas/Acinetobacter*=40%).

Revised text as it appears in text (Lines 70-75): “ β -lactamases, particularly ESBLs, have been identified as major drivers of resistance in animals in Ecuador, representing some of the most clinically relevant ARGs circulating in the region. Recent metagenomic and genomic studies demonstrate that β -lactamase ARGs such as CTX-M are not only widespread but also frequently associated with mobile genetic elements, increasing their potential for horizontal gene transfer (HGT) across hosts and environments.”

Supplemental Figure S2.

- Birth Mode and Feeding Information: Please report the number of infants delivered vaginally and the number who were exclusively breastfed. Breastfeeding influences the microbiome and the resistome. This information is crucial to contextualize resistome development.

Response: We thank the reviewer for this suggestion. We have now included delivery mode information in **Table 1**. 77.4% of infants were delivered vaginally and 22.6% by C-section. Regarding breastfeeding, most infants were exclusively breastfed during the first month of life (72%), but this proportion decreased with age. Because breastfeeding was strongly correlated with child age (as indicated by variance inflation factor [VIF] analysis), we removed breastfeeding as a covariate from the children’s models to avoid collinearity.

Revised text as it appears in text: Table 1

- Mother–Infant Pairs: Clarify the number of complete mother–infant pairs included in the study. In Section 3.1, households should be described based on cohabitating mother–infant pairs rather than treating mothers and infants separately. What is the size of the family? did you check the impact of caregivers, siblings and other members living together?

Response: We clarified that our study included 53 mother–infant pairs, with infants sampled at one month of age. The exact selection of samples is presented in **Table S1**. The median household size was 5 (range: 2–11), and models are now adjusted for household size. This has been clarified in the methods section.

Revised text as it appears in text (Lines: 202-203): “Our study included 53 mother–infant pairs, with infants sampled at one month of age. The exact selection of samples is presented in Table S1.” and Supplementary Table S1.

All linear model results have been updated (e.g., Figures 1, 3, 4).

- Lines 400–411: Please, a comparison between mother and infants will be interesting. Please, include a figure that compares mothers and infants to complement this section.

Response: We appreciate the reviewer's suggestion. We included a new figure (**Figure 2**) that directly compares mothers and children. Panel A shows an NMDS ordination based on the abundance of clinically relevant ARGs, highlighting clear differences in ARG composition between mothers and children. Panel B complements this analysis by comparing both the prevalence and relative abundance of *E. coli* and *K. pneumoniae* across age groups in children and mothers.

Revised text as it appears in text: Figure 2

Lines 389-391: "In addition, Non-metric multidimensional scaling (NMDS) ordination indicates differences in the composition of clinically relevant ARGs between mothers and children (Figure 2A)."

Lines 456-457: "One week old children had the lowest prevalence of *E. coli* (75.5%) but highest concentration in positive samples (Figure 2B)."

- Did the authors analysed the pets microbiota to confirm the origin/transfer of ARGs to the families?

Response: We did not sequence animal fecal samples in this study, so we are unable to confirm direct ARG transfer from pets. However, in a related sub-study, we collected ESBL-producing bacteria isolates from animals in a subset of the same households. These isolates showed resistance to third-generation cephalosporins, overlapping with ARG classes identified in the human gut resistomes. While this does not confirm direct transmission, it supports the plausibility that household animals act as reservoirs of clinically relevant ARGs. We added animal feces culture data to the supporting information, clarified this limitation in the discussion, and proposed future inclusion of animal sampling.

Revised text as it appears in text:

Methods Section 2.3: *Sample and data collection*

Lines 191-197: "For the purposes of verifying animal exposure as a source of antimicrobial resistant bacteria in our study setting, we collected ESBL-producing bacterial isolates from animals in a subset of the same households. As part of the EcoMid AnEx sub-study, 47 fecal samples were obtained from dogs, chickens, pigs, cows, and ducks and cultured to screen for ESBL-producing bacteria. Presumptive *E. coli* isolates were subsequently subjected to antimicrobial susceptibility testing following CLSI guidelines. Further methodological details are provided in the Supplementary Information. We note that animal fecal samples were not included in the metagenomic analysis."

Results Section 3.1: *Overview of households included in this study and environmental exposures*

Lines 366-369: “Examining animal feces as a source of AMR exposure, ESBL-producing bacteria were present across several animal species (e.g. dogs, chicken, cows, pigs and ducks) and ESBL E. coli was detected in 88% of chicken fecal samples (Figure S2). Many of these isolates were resistant to third-generation cephalosporins (e.g., ceftriaxone, ceftazidime) (Figures S3).”

Discussion Section:

Lines 560-565: “While we did not sequence animal fecal microbiomes in this study, ESBL-producing bacteria were prevalent across several animal species. Many of E.coli isolates were resistant to third-generation cephalosporins (e.g., ceftriaxone, ceftazidime), overlapping with ARG classes identified in the human gut resistome. Although we cannot directly demonstrate transmission between animals and humans in this study, these results support the plausibility that animals act as reservoirs of clinically relevant ARGs in these communities.”

- Lines 461–469: It would enhance clarity to display all MAGs and their predicted resistance profiles in a phylogenetic tree.

Response: We appreciate the suggestion. However, we did not generate a phylogenetic tree for the MAGs because the analysis would not be robust in our case. Four ESKAPEE MAGs were recovered, which provides very limited phylogenetic resolution and results in a tree with minimal interpretative value. We are also cautious that presenting a tree with such few MAGs could lead to overinterpretation of the limited branching patterns as evidence of evolutionary novelty or transmission dynamics. Importantly, the focus of our manuscript is on resistome dynamics and the prevalence/abundance of clinically relevant pathogens, rather than on reconstructing their phylogenetic relationships. For these reasons, we believe that a phylogenetic tree would not add clarity or strength to our findings, and we instead provide detailed information on the recovered MAGs and their resistance profiles in the text.

Revised text as it appears in text: No changes made.

- Discussion – Opening Summary: Begin the Discussion section with a succinct summary of the study’s key findings.

Response: We thank the reviewer for this helpful suggestion. We have revised the beginning of the Discussion to include a succinct summary of the study’s key findings.

Revised text as it appears in text (Lines 548-553): “In this study, we examined how animal exposure and household water and sanitation conditions influence the gut resistome and microbiome in mothers and children. We found stronger associations between animal exposure and resistance patterns in mothers compared to children, while improved water and sanitation were generally linked to lower ARG abundance. Notably, we detected clinically relevant β -lactamase genes, underscoring the public health significance of

these findings and the potential role of household exposures in shaping the human resistome.

- Line 481–482: This statement appears to contradict the findings described in lines 360–362. Please revise to ensure internal consistency.

Response: We appreciate the reviewer pointing out this potential inconsistency. We clarified the text to more accurately reflect our findings. In mothers, we observed that low and medium levels of animal exposure were associated with significantly lower ARG diversity compared with high exposure. However, the number of clinically relevant ARGs in mothers with zero exposure appeared more similar to those with high exposure, which we interpret cautiously given the difficulty of defining “zero exposure” in these communities where indirect exposure via environmental reservoirs is likely. In children, the patterns were more variable, and no statistically significant differences were observed across exposure levels. We revised the discussion to better distinguish between these nuanced patterns, ensuring consistency with the results presented.

Revised text as it appears in text (Lines 554-558): “We found evidence that exposure to animals may be associated with increased number of clinically relevant ARGs and taxonomic diversity in adults (mothers), while zero exposure appeared more similar to high exposure, likely reflecting indirect exposure routes in these communities. In children, however, no clear or statistically significant associations with animal exposure were observed.

- Line 578–580: The current phrasing suggests that the number of recovered MAGs directly reflects the relative abundance of microbes in the microbiome. However, this is misleading, as MAGs recovery is more strongly influenced by sequencing depth and assembly efficiency than by actual microbial abundance. Please revise this statement.

Response: We thank the reviewer for this important clarification. We agree that the number of recovered MAGs should not be interpreted as a direct reflection of microbial abundance. We have revised the text accordingly. In the revised version, we clarify that *E. coli* and *K. pneumoniae* were not only the most frequently detected ESKAPEE pathogens but also the most abundant across samples, while removing any suggestion that MAG recovery itself reflects abundance. The updated text is now provided in

Revised text as it appears in text (Lines 669-674): “*E. coli* and *K. pneumoniae* were the most frequently detected ESKAPEE pathogens in the metagenomes and showed the highest relative abundances across samples. MAGs and StrainGE-identified strains of these species were found in most mother and child samples. In contrast, *A. baumannii* MAGs were notably absent in all mother samples. The observed differences in prevalence and relative abundance of ESKAPEE pathogens may reflect distinct environmental reservoirs or differences in host susceptibility among age groups.

- The comparison of StrainGE results and MAGs is not included in the Results section. Please include or explicitly justify its omission.

Response: We thank the reviewer for this helpful suggestion. In response, we generated a new figure (**Figure S2**) directly comparing ESKAPEE pathogen relative abundances obtained using MAG-based approaches and StrainGE. In addition to the direct comparison between methods, we also note that the conclusions (associations with animal exposure) do not change when we investigate StrainGE determined relative abundance of *E. coli* and *K. pneumoniae* and MAG coverage/GE of *E. coli* and *K. pneumoniae*.

Revised text as it appears in text (Lines 277-279): “To evaluate the consistency of ESKAPEE pathogen detection, we compared MAG-based coverage to relative abundances obtained using StrainGE by performing linear regressions.

Lines 526-530: “We also observed that the coverage estimated from MAGs (coverage in TAD80/GE) and relative abundance from StrainGE were highly correlated for all ESKAPEE pathogens, including *E. coli* ($R = 0.8$, $p < 0.05$) and *K. pneumoniae* ($R = 0.84$, $p < 0.05$) (Figure S4). Both approaches identified *E. coli* and *K. pneumoniae* as the most abundant and frequently detected pathogens across mothers and children.”

Minor comments

- Consistently use a single scheme for tertile classification—either "high, medium, low" or "upper, middle, lower"—throughout the text and figures

Response: We standardized all text, tables, and figures to use “**low, medium, high**” exclusively.

- Panels C and D of Figure 1 and Panel B of Figure 3 are not mentioned in the main text. Please ensure all figure panels are explicitly cited and discussed.

Response: We now reference **Panels C & D of Figure 1** and **Panel B of Figure 3** in the main text.

- Line 390: Include a reference to the appropriate panel letter of Figure 2.

Response: We thank the reviewer for pointing this out. We have clarified the text to specify the appropriate panel letters and now refer to **Figures 3A and 3B**.

- Figure 4 Caption (Line 436): Do not re-abbreviate "antibiotic resistance genes" (ARGs) in the figure caption if the abbreviation has already been defined.

Response: We removed redundant re-abbreviation of “ARGs” in Figure 4 caption now **Figure 5**.

- Line 454: specify the age of the children you are comparing here.

Response: We appreciate the reviewer's suggestion. In this sentence, we are referring to all children across the sampled age groups compared to mothers. We have revised the text to make this clearer.

Revised text as it appears in text (Lines 517-519): "ESBL genes were consistently found in mothers and children across age groups, although the majority of ARG containing contigs were identified in children overall compared to mothers (Figure 4A)"

- Indicate the sample size (n) for each group in the relevant figures for better interpretability.

Response: We have added "n=" annotations per group in Figures 1, 2, 3, and 4.

- Line 556: The claim made here is too strong, as it is based solely on b-lactamases without comparison to other ARG classes. Please rephrase to reflect the scope of your data.

Response: We agree with the reviewer and have rephrased the sentence to more accurately reflect the scope of our data.

Revised text as it appears in text (Lines 649-651): "The detection of clinically relevant beta-lactamase genes, including *blaCTX-M*, *blaOXA*, *blaSHV*, and *blaTEM*, across all age groups highlights the presence and persistence of these resistance determinants in the study population."

Reviewer #2 (Remarks to the Author):

With this study, Cotto et al. explored the relationships between environmental exposures (e.g., animal contact, water and sanitation access) and the human gut resistome by leveraging metagenomic sequencing and statistical modeling in a large cohort of mothers and children in rural Ecuador. Using a combination of shotgun metagenomic sequencing, strain-level profiling, and generalized estimating equations, they assess how animal exposure and WASH (water, sanitation, and hygiene) infrastructure impact gut microbial diversity, ARG (antibiotic resistance gene) abundance and diversity, and the presence of clinically relevant ESKAPEE pathogens. They found that (1) mothers with higher animal exposure showed greater taxonomic diversity and marginally higher ARG diversity; (2) improved WASH infrastructure (specifically combined access to piped water and contained sewage) was associated with significantly lower ARG abundance and richness; and (3) ARGs, including beta-lactamases, were frequently detected in ESKAPEE pathogens such as *E. coli* and *K. pneumoniae*, often on plasmid-borne contigs. This is an important and timely study, and the authors successfully conducted detailed metagenomic work in a challenging field setting, particularly involving sampling of both adults and young children. The integration of environmental context with high-resolution resistome data is a valuable contribution to the growing literature on antimicrobial resistance in low- and middle-income settings. However, despite the study's strengths in large-scale data collection and bioinformatic analysis, several important mechanistic gaps and interpretive

assumptions remain unresolved. Below are some specific concerns and suggestions for improvement.

Response: We thank the reviewer for their comments. We have addressed the concerns as specified below.

1. The authors suggest that increased ARG diversity in mothers with high animal exposure may result from microbial spillovers from animals. However, this claim remains speculative, as the study does not provide taxonomic resolution of ARG carriers nor track gene flow between animal microbiota and the human gut. No taxonomic enrichment analyses, ARG-host assignment, or co-occurrence networks were performed to support this conclusion. Furthermore, animal fecal microbiomes were not sequenced, limiting source attributions. The authors may consider performing metagenomic co-assembly and genome binning to link ARGs to host taxa, or may use co-occurrence or network-based analysis to identify bacterial lineages enriched in high-exposure groups. In addition, sequencing animal microbiomes and comparing shared ARG profiles, although they have mentioned, need to have animal fecal data.

Response: While we did not sequence animal feces in this study, we did isolate ESBL-producing *bacteria* from animals in a subset of households in a sub-study. *E. coli* isolates showed resistance to third-generation cephalosporins (e.g., ceftriaxone, ceftazidime), overlapping with ARG classes identified in the human gut resistome. Although we cannot directly demonstrate gene transfer between animals and humans in this study, these results support the plausibility that animals act as reservoirs of clinically relevant ARGs in the study communities. These analyses support the hypothesis that animal-associated bacteria (e.g., Enterobacteriaceae) carry ARGs that may contribute to the resistome shifts we observed. We have added these culture data to the supporting information and mention of the results in the discussion.

Revised text as it appears in text

Methods Section 2.3: *Sample and data collection*

Lines 191-197: “For the purposes of verifying animal exposure as a source of antimicrobial resistant bacteria in our study setting, we collected ESBL-producing bacterial isolates from animals in a subset of the same households. As part of the EcoMid AnEx sub-study, 47 fecal samples were obtained from dogs, chickens, pigs, cows, and ducks and cultured to screen for ESBL-producing bacteria. Presumptive *E. coli* isolates were subsequently subjected to antimicrobial susceptibility testing following CLSI guidelines. Further methodological details are provided in the Supplementary Information. We note that animal fecal samples were not included in the metagenomic analysis.”

Results Section 3.1: *Overview of households included in this study and environmental exposures*

Lines 366-369: “Examining animal feces as a source of AMR exposure, ESBL-producing bacteria were present across several animal species (e.g. dogs, chicken, cows, pigs and ducks) and ESBL *E. coli* was detected in 88% of chicken fecal samples (Figure S2). Many of these isolates were resistant to third-generation cephalosporins (e.g., ceftriaxone, ceftazidime) (Figures S3).”

Discussion Section:

Lines 560-565: “While we did not sequence animal fecal microbiomes in this study, ESBL-producing bacteria were prevalent across several animal species. Many of *E.coli* isolates were resistant to third-generation cephalosporins (e.g., ceftriaxone, ceftazidime), overlapping with ARG classes identified in the human gut resistome. Although we cannot directly demonstrate transmission between animals and humans in this study, these results support the plausibility that animals act as reservoirs of clinically relevant ARGs in these communities.”

2. While the reduction in ARG richness and abundance in mothers with CS&PW infrastructure is statistically robust, the mechanistic interpretation that WASH reduces ARGs via lower antibiotic use and fewer HGT events is unsupported by direct evidence. No data on antibiotic consumption was collected, nor were HGT rates or mobile genetic elements explicitly analyzed.

Response: We agree that this inference is indirect. We have revised the discussion to remove any speculative causal wording, clarify that while epidemiological studies link higher quality WASH to reduced infection burden (and antibiotic use), we did not report antibiotic consumption or HGT rates. We also clarify that our study is epidemiological and not mechanistic. We now delineate antibiotic use or reduced horizontal transfer as plausible but unmeasured in this work. We also note that collecting accurate antibiotic consumption data in both humans and animals in this setting, where unregulated use is common, is challenging. For humans, antibiotics are often mistaken as vitamins and for animals, animal feed or other veterinary products often have antibiotics without owners knowing².

Revised text as it appears in text (Lines 614-617): “While we do not report data on antibiotic consumption or directly assess HGT, prior work has shown that improved WASH conditions can reduce the incidence of infections, which may in turn lower antibiotic use and limit opportunities for ARG dissemination.”

3. The findings for children differed notably from those for mothers, with animal exposure showing no consistent trends and WASH having weaker or nonsignificant associations. This divergence was not deeply discussed or contextualized. The authors should address why children’s resistome patterns do not mirror their mothers’. Are children less exposed due to behavioral factors? Does gut microbial development or antibiotic history play a role? Including hypotheses or preliminary comparisons would improve clarity.

Response: We thank the reviewer for this helpful comment. We expanded the discussion to address potential explanations for the divergence between mothers

and children. Specifically, we now note four factors: (i) the difficulty of defining “zero exposure” in households with animals; (ii) the dynamic nature of the infant gut microbiome; (iii) the behavioral and environmental differences and (iv) differences in antibiotic exposure and recovery between adults and children. These additions are now included in the Discussion.

Revised text as it appears in text (Lines 591–608): “Several factors may explain differences in associations between mothers and children. First, defining “zero exposure” in households with animals is challenging, which could obscure trends. Second, the dynamic nature of the infant gut microbiome, influenced by breastfeeding, dietary shifts, and infections, likely plays a central role.^{84,85} For example, breastfeeding promotes the growth of beneficial bacteria like *Bifidobacterium*,⁸⁶ while introducing solid foods leads to changes in the composition of the gut microbiome.^{86,87} Third, behavioral and environmental differences may result in different types of animal contact for infants compared to mothers. For instance, mothers are more likely to handle animal husbandry tasks such as feeding, cleaning, or managing waste, which increases the chance of direct exposure to animal-associated microbes and ARGs.⁸⁸ Infants may be intentionally shielded from animals by caregivers to reduce perceived health risks and direct contact opportunities.⁸⁹ Finally, differences in antibiotic exposure and recovery may also explain the divergence between mothers and children. Adult gut microbiomes typically recover from a single antibiotic exposure within about two weeks, whereas children often exhibit less stability, with recovery sometimes extending beyond one month.⁹⁰ Such reduced resilience in early life may contribute to the distinct resistome patterns observed in children compared to mothers.”

4. The detection of clinically relevant ARGs in ESKAPEE MAGs is important, but this study does not demonstrate actual transfer of ARGs into these pathogens during the study period. Many of the ARGs reported (e.g., *bla*CTX-M, *bla*TEM) are well known and widely distributed. Without longitudinal sampling, full plasmid resolution (e.g., via long-read sequencing), or direct functional assays, it is not possible to infer recent horizontal gene transfer or de novo ARG acquisition.

Response: We appreciate the reviewer’s thoughtful comment highlighting the limitations of interpreting plasmid-borne ARGs in our dataset. We have revised the discussion to clarify that our findings demonstrate the presence of plasmid-associated ARGs in ESKAPEE MAGs but do not provide direct evidence of recent HGT or ARG acquisition. We now emphasize that the identification of plasmid-borne ARGs is consistent with their known role in resistance dissemination, while noting that further long-read and functional studies would be necessary to assess mobility and transfer dynamics.

Revised text as it appears in text (Lines 658-668): “In our dataset, many clinically relevant ARGs detected in ESKAPEE MAGs were carried on plasmids, including *bla*CTX-M-15, *bla*OXA-471, and *bla*SHV-40, whereas others such as *bla*SHV-1 and *bla*OXA-1 were primarily chromosomal. The presence of

plasmid-borne ARGs is consistent with the well-documented role of mobile genetic elements in facilitating the dissemination of resistance traits across bacterial populations. Plasmids have been widely recognized as vectors of ESBL genes among *Enterobacteriaceae*, contributing to the global spread of multidrug resistance. While our findings highlight that plasmid-associated ARGs are present within clinically important pathogens in this community, we did not directly assess plasmid mobility, recent HGT events, or de novo ARG acquisition. Additional long-read sequencing and functional assays would be required to determine the dynamics of ARG exchange between bacterial hosts.”

5. Moreover, the study would benefit from complementary culture-based validation. Given that *E. coli*, *K. pneumoniae*, and other ESKAPEE pathogens are routinely culturable from stool, the authors could have attempted to isolate these strains and screen them phenotypically or genomically for ARG content. Cultured those isolates would be better for direct correlation of ARGs with host genotype and phenotype. In short, in addition to metagenomic inference, the author should consider strengthening the study with isolate-based approaches — particularly for common gut pathogens that are straightforward to culture and characterize.

Response: We thank the reviewer for this valuable suggestion. We agree that complementary culture-based validation would provide important insights into linking ARGs with host genotype and phenotype, particularly for *E. coli* and *K. pneumoniae*, which are readily culturable from stool. While our current study focused on shotgun metagenomic approaches to broadly characterize the gut resistome, we acknowledge the added value of integrating isolate-based analyses. We noted this limitation in the discussion and emphasized that future studies incorporating both metagenomics and culture-based methods will be crucial to validate ARG carriage, explore phenotypic resistance, and directly assess the relationship between ARGs, host background, and horizontal gene transfer potential. While it is not logistically or financially feasible to culture and sequence isolates from the human fecal samples, we did conduct a sub-study on isolates from animal feces and added those results.

Revised text as it appears in text (Lines 722-729): “Moreover, our study relied solely on metagenomic short-read sequencing, which limits the ability to directly link ARGs with their bacterial hosts or confirm phenotypic resistance. Culture-based approaches, particularly for *E. coli*, *K. pneumoniae*, and other readily culturable gut pathogens, would provide important complementary data by enabling genomic and phenotypic characterization of isolates and exploring HGT potential. Future studies incorporating direct sampling of both pets and livestock, detailed antibiotic usage data, and complementary isolate-based approaches will enhance our understanding of the pathways and mechanisms of antimicrobial resistance dissemination”

Figures S2 and S3.

Reviewer #3 (Remarks to the Author):

Response: We thank you for co-reviewing this manuscript.

Reviewer #4 (Remarks to the Author):

Response: We thank you for co-reviewing this manuscript.

REFERENCES

1. Zhang, A.-N. *et al.* An omics-based framework for assessing the health risk of antimicrobial resistance genes. *Nat Commun* 12, 4765 (2021).
2. Braykov, N. P. *et al.* Antibiotic Resistance in Animal and Environmental Samples Associated with Small-Scale Poultry Farming in Northwestern Ecuador. *mSphere* 1, (2016).

REVIEWERS' COMMENTS

Reviewer #1 (Remarks to the Author):

Authors have covered my questions and concerns. Paper is improved.

Response: We thank the reviewer for their comments to improve the paper.

Reviewer #2 (Remarks to the Author):

Thank you for the revisions. The manuscript is clearer, several claims are tempered, and the added culture snapshot is informative. However, there's no substantive evidentiary advance (e.g., taxonomic enrichment, co-occurrence/network analysis, robust ARG–host linkage, antibiotic/MGE data, or human-isolate validation), so our core comments remain not fully addressed.

Response: We thank the reviewer for their helpful feedback in improving our manuscript. Without a substantial amount of funding, it is not possible to culture the human fecal samples or conduct additional methods for ARG and host linking (e.g., HiC). We rigorously analyzed our metagenomic data by assembling contigs and metagenome-assembled genomes. In addition, this was an epidemiological study, and additional studies would need to be conducted to investigate mechanisms as the reviewer suggests.

Reviewer #3 (Remarks to the Author):

Response: We thank the reviewer for contributing to the review.

Reviewer #4 (Remarks to the Author):

Response: We thank the reviewer for contributing to the review.